# A Multi-site Passive Approach for Studying the Emissions and Evolution of Smoke from Prescribed Fires

Rime El Asmar[1], Zongrun Li[2], David J. Tanner[1], Yongtao Hu[2], Susan O'Neill[3], L. Gregory Huey[1], M. Talat Odman[2], Rodney J. Weber[1]

[1]School of Earth and Atmospheric Sciences, Georgia Institute of Technology, Atlanta, 30331, USA.
[2]School of Civil and Environmental Engineering, Georgia Institute of Technology, Atlanta, 30331, USA.
[3]USDA Forest Service, Pacific Northwest Research Station, 400 North 34th Street, Suite 201, Seattle, WA 98103, USA.

*Correspondence to*: Rodney J. Weber (*rweber@eas.gatech.edu*)

**Abstract.** We conducted a two-year study utilizing a network of fixed sites with sampling throughout an extended prescribed burning period to characterize the emissions and evolution of smoke from silvicultural prescribed burning at a military base in the southeastern US. The measurement approach and an assessment of instrument performance is described. Smoke sources, including those within and off the base, are identified, and plume ages are determined to quantify emissions and study the evolution of smoke $PM_{2.5}$ mass, black carbon (BC), and brown carbon (BrC). Over the 2021 and 2022 prescribed burning seasons (nominally January to May), we identified 64 smoke events based on high levels of $PM_{2.5}$ mass, BC, BrC, and carbon monoxide (CO), of which 61 were linked to a specific burning area. Smoke transport times were estimated using the mean wind speed along with the distance between fire and measurement site, and with HYSPLIT back trajectories. $PM_{2.5}$ emission ratios based on $\Delta PM_{2.5}$ mass/$\Delta CO$ for fresh smoke (age $\leq$ 1 hour) ranged between 0.04 and 0.18 $\mu g\ m^{-3}\ ppb^{-1}$ with a mean of 0.117 $\mu g\ m^{-3}\ ppb^{-1}$ (median of 0.121 $\mu g\ m^{-3}\ ppb^{-1}$). Both the mean emission ratio and variability were similar to findings from other prescribed fire studies, but lower than wildfires. Mean emission ratios of BC and BrC were 0.014 $\mu g\ m^{-3}\ ppb^{-1}$ and 0.442 $Mm^{-1}\ ppb^{-1}$ respectively. Ozone enhancements ($\Delta O_3$) were always observed in plumes detected in the afternoon. $\Delta PM_{2.5}$ mass/$\Delta CO$ was observed to increase with plume age in all ozone enhanced plumes suggesting photochemical secondary aerosol formation. In contrast, $\Delta BrC/\Delta CO$ was not found to vary with plume ages less than 8 hours during photochemically active periods.

## 1. Introduction

Large and intense wildfires have been increasing over the past few decades and their emissions are a critical concern (Singleton et al., 2019; Jaffe et al., 2020). Fire is also an essential ecological process and prescribed burning, which is the act of starting controlled fires for specific purposes, is an important tool for restoration of ecosystems, land management, and reducing fuel to prevent destructive wildfires (Kelp et al., 2023). Prescribed fires are typically conducted during favorable conditions associated with the fuel type and amount, soil moisture, and meteorology. For example, in 2018, the United States Department of Agriculture (USDA) Forest Service indicated a high risk of hazardous wildfires over approximately 234 million acres (~ 95 million ha) of forest lands in the US (Wyden and Manchin, 2020). However, prescribed fires were conducted over approximately 8.5 million forestry/rangeland acres (3.4 million ha) in 2018 (Melvin, 2020). The southeastern US has a long history of using prescribed fires (Melvin, 2021). For example, in 2017, 7.6 million acres (3 million ha) out of the 11.3 million acres (4.6 million ha) burned nationally were in the southeast (Melvin, 2018). Florida and Georgia each exceeded 1 million acres (0.4 million ha) burned annually (Melvin, 2018). Recognizing the need to mitigate the size and severity of wildfires, prescribed burning is anticipated to increase in the coming years (USDA, 2022).

While prescribed burning can be performed under favorable weather conditions, it can still contribute to serious local and regional air pollution as it is a source of primary and secondary air pollutants (Lee et al., 2008). Like other types of biomass burning, prescribed burning releases large amounts of particulate matter, CO, and inorganic and organic compounds (Lee et al., 2005), which have negative effects on health and visibility (Bell, 2004; Huang et al., 2019). Particularly in the southeastern US, prescribed burning was significantly associated with high $PM_{2.5}$ levels (Afrin and Garcia-Menendez, 2020; Larkin et al., 2020). Prescribed fires are often conducted at urban-rural interfaces creating a buffer zone to prevent the spread of wildfires towards the built environment. However, this means that the planned fires often occur closer to populated areas, and potentially lead to high population exposure due to this proximity. Although prescribed fires generally produce less pollutants by consuming less fuel per area burned than wildfires, the population health costs can be substantially higher for prescribed fires due to burning near higher population densities (Borchers-Arriagada et al., 2021).

Both wildfires and prescribed fires emit a large variety of gases and particulates (Liu et al., 2017b; Burling et al., 2011; Gkatzelis et al., 2024; Permar et al., 2021; Travis et al., 2023). Gases include nitrogen oxides and volatile organic compounds that can form ozone and secondary particulate matter. Hazardous air pollutants are also produced but they may be less detrimental to exposed populations than particulates (O'Dell et al., 2022). $PM_{2.5}$, (particulate matter with aerodynamic diameter of 2.5 micrometers or smaller), is directly emitted as primary particles and also formed from condensation of emitted gases and their oxidation products (Liu et al., 2016; May et al., 2014). While secondary organic aerosol (SOA) can be a significant component of aged biomass burning $PM_{2.5}$, its contribution changes depending on emissions and atmospheric conditions. Additionally, the volatile nature of primary and secondary components of $PM_{2.5}$ can lead to evaporation and a net loss in mass as the plume ages. $PM_{2.5}$ exposure has been linked in many epidemiological studies to serious health problems such as respiratory, cardiovascular, and neurological diseases, as well as increased risk of adverse birth outcomes (Liu et al., 2015; Reid et al., 2016; Naeher

et al., 2007; Yu et al., 2023; Xi et al., 2020; Garcia et al., 2023). Given their significant impact on the environment and health, satellite, airborne, or ground-based studies of smoke emissions have been extensively conducted.

Detection and characterization of wildland fires is an important step towards assessing their impacts. Remote sensing via satellites can detect wildland fires by thermal anomalies (Kuenzer et al., 2008) or vegetation changes (Mildrexler et al., 2007). While satellite-based approaches offer valuable insights (Martinsson et al., 2022; Ichoku and Kaufman, 2005; Christopher et al., 1998), challenges such as cloud cover, spatial resolution limitations, and the complex nature of fire emissions can hinder accurate detection and quantification of fire impacts, especially for lower-intensity fires like prescribed burns (Liu et al., 2019; Wang et al., 2018; Martin et al., 2018). Therefore, factors like Fire Radiative Power (FRP), burned area estimation, and fuel consumption modeling are often integrated into fire monitoring systems (Li et al., 2020; Nguyen and Wooster, 2020).

Aircraft (fixed wing and helicopters) and more recently drones are commonly used in airborne studies of wildland fires (Decker et al., 2021b; Cubison et al., 2011; Aurell and Gullett, 2024) and have been deployed for prescribed burning studies (Yokelson et al., 1999; May et al., 2014; Pratt et al., 2011; Aurell et al., 2021). Airborne studies provide high spatial resolution data that are often used to assess evolution of smoke properties by measurements at various downwind distances, however, it is non-continuous, and can miss certain aspects of smoke emissions, such as longer-term smoldering, especially at night (Burling et al., 2011). Employing a combination of airborne and ground-based measurements can be beneficial in providing a comprehensive view of the plume (Burling et al., 2011; Akagi et al., 2014; Yokelson et al., 2013; Strand et al., 2016).

In ground-based studies, mobile labs may capture dynamic air quality patterns and to some extent assess spatial variability of species in plumes and their changes with plume age (Levy et al., 2014; Fiddler et al., 2024; Lee et al., 2023). However, they are usually limited in space and instrumentation capacity, such as filter samples collected only during stationary measurements (Warneke et al., 2023). Interferences from the power source, vibration and speed changes during transportation can affect instrument stability and performance leading to inaccurate measurements or limiting the type of instruments that can be used. Attempting to track wildland smoke plumes can be challenging due to unpredictable winds and dispersion conditions combined with access limitations. For example, Burling et al. reports successfully sampling smoke from 2 out of 14 prescribed fires using a battery powered mobile FTIR system (Burling et al., 2011).

Fixed ground-based monitoring stations equipped with various instruments provide continuous, localized measurements for short or long-term monitoring for studies assessing diurnal, seasonal, and long-term trends in air pollution. Multiple sites provide spatial coverage within a region. A variety of highly sensitive instruments can be deployed, ensuring accurate and precise measurements of various pollutants that can be compared with air quality data across different locations for regional assessments (Strand et al., 2016; Warneke et al., 2023). The importance of pre-existing fixed monitoring sites lies in their ability to capture wildfire smoke events that can occur at any time (Selimovic et al., 2019; Jaffe et al., 2022). These sites often include regulatory monitoring stations, which are highly valuable for studying local and regional smoke impacts over both short and long-term periods. For example, Jaffe et al. used $PM_{2.5}$ and CO observations from a regulatory monitoring site in Sparks, NV, collected from May to September between 2018 and 2021, as indicators of wildfire smoke in urban areas (Jaffe et al., 2022). Investigating emissions

and evolution of prescribed fires based on fixed sites is not as common, and there are limitations with this approach,
but also some advantages.
Here, we present results from a two-year study utilizing fixed monitoring stations and continuous sampling
in a region of active prescribed burning at Fort Moore in central Georgia, USA. The observations are analyzed to
identify smoke plumes and determine their sources, such as those set within the Fort or from burning in surrounding
areas. We also use these data to estimate the age of the smoke detected to determine emission ratios and changes with
plume age of $PM_{2.5}$ mass, BC, and BrC and their variability. Not all smoke from the prescribed fires set within the
Fort are detected so the overall impact of all fires on regional air quality cannot be determined and is better addressed
by a model simulation. Instead, our goal is to sample multiple smoke events so that an analysis of the data will provide
a robust characterization of smoke from prescribed burning within the Fort and in the region and sufficient data to
evaluate ground-level pollutant concentrations predicted by "smoke" models in prescribed fire simulations. Our
concentration data cover measurements over a large range of distances from the burn plots. Fresh plume measurements
with ages less than 1 hour can be used in evaluating the predictions of local scale models such as the Wildland urban
interface Fire Dynamics Simulator (WFDS) (Mell et al., 2007) and the QUIC-Fire (Linn et al., 2020). They can also
be used in evaluating the emissions and plume-rise parameterizations of larger scale models like the BlueSky
framework (Larkin et al., 2009). Additionally, more aged smoke measurements can be used to test the predictions of
downwind concentrations in coupled fire-atmosphere models such as WRF-SFIRE (Mandel et al., 2011) as well as
chemical transport models like the Community Multiscale Air Quality (CMAQ) model (Appel et al., 2021), when they
are equipped with fire plume parameterizations. In the following sections, we describe the methodology, data analysis
approach, case studies of various detected or missed smoke plumes so that attribution of smoke from fires within the
Fort can be assessed. Findings on emission estimates of $PM_{2.5}$ mass, BC, and BrC and their evolution are compared
to other prescribed and wildfire studies. These findings can help to assess the impact of prescribed burns by a specific
entity or organization on a variety of public health and policy issues.
**2. Method**
**2.1. Site description**
Prescribed burning at Fort Moore Army Base, (formerly Fort Benning), in west central Georgia, United
States, was studied during March through May of 2021 and February through May of 2022. Since 1981, prescribed
burning has been used as a land management tool at the 182,000 acres (~ 74, 000 ha) military base, of which 145,000
acres (~ 59,000 ha) are forested lands. Vegetation is characterized by pine-dominated uplands and hardwood-
dominated bottomlands, with the dominant tree species being longleaf pine and white oak, respectively. Small
wildfires ignited during military training exercises also occur at the base and the land managers have been recording
data on both prescribed fires and wildfires since the 1980s. Prescribed burning at the Fort has been effective; it has
reduced the frequency of wildfires from ~ 300-500 wildfires/year in the early 1980s to less than 100 wildfires/year in
the mid-1990s. During this period the prescribed fire burnt area changed from ~7,500 acres (~ 3,000 ha) in 1981 to ~
12,000 acres (~ 5,000 ha) in 1992. Currently, 30,000 woodland acres (~ 12,000 ha) are burned annually using

controlled fires, with a future planned burning of 45,000 acres (~ 18,000 ha) annually. Prescribed burning on the Fort is also used for ecological objectives, such as restoring the longleaf pine forest and creating and maintaining habitat for red-cockaded woodpeckers. Prescribed burning occurs from December through May when there is sufficient but not excessive rainfall, and suitable temperatures and wind conditions to burn deadwood, brush, and low-growing vegetation accumulating on the forest floor. The area of the base is divided into 332 burn units that range in size from 100 to 1,800 acres (~ 40 to 728 ha) and are burnt alternately every two to three years.

**2.2. Measuring sites**

One instrumented research trailer (7'W x 18'L x 6.5'H) was deployed in the 2021 burning season (March 18, 2021 to May 15, 2021), and successively trailers (6'W x 12'L x 7'H) were added in 2022 (February 11, 2022 to May 18, 2022) reaching a total of five trailers located at different sites throughout the Fort. In 2021, the one trailer operated at the same location until it was moved on April 26, 2021 to a new site for the remaining season as expected burning regions at the Fort changed. The trailers sampled continuously, except during periods of power loss or technical issues. The locations of trailers, shown in Fig. 1, were chosen based on power availability, prevailing wind, and burning plans set prior to the burning season.

**2.3. Instrumentation**

To characterize the prescribed fire smoke, the trailers were equipped with several instruments selected based on factors such as availability, ability for extended stand-alone operation, and their significance to the study. All sampling was done through inlets nominally 4 m above ground level and 1.5 m above the trailer roof. Measurements included, carbon monoxide (CO), nitrogen oxides (NO, $NO_2$, $NO_x$), ozone ($O_3$), $PM_{2.5}$ mass concentration and black carbon (BC) concentration and brown carbon (BrC) light absorption coefficients. Carbon monoxide serves as a standard tracer for combustion sources in atmospheric chemistry studies since it is a relatively long-lived species, with a typical lifetime of ~ 1 month, emitted during incomplete combustion and used as a tracer of smoke movement and dispersion (Forrister et al., 2015; Liu et al., 2016). Other forms of incomplete combustion emissions (e.g., mobile sources) and oxidation of VOCs are also CO sources. CO mixing ratios were measured by IR analyzers (Thermo Fisher Scientific Inc, model 48C, Franklin, MA) with a lower detection limit (LOD) of 0.04 ppm at an averaging time of 390 seconds. The measurements alternated between blank and ambient measurements every 195 seconds. The blanks were determined with a custom-built CO scrubber made of 0.5 % Pd on alumina catalyst heated to 180 °C (Parrish et al., 1994), which oxidizes CO to $CO_2$. Calibration of CO analyzers was performed at 2.2 ppm concentration before and after each field study using a 100 ppm CO in air standard purchased from nexAir (Memphis, TN).

$O_3$ was measured using an ultraviolet (UV) photometric analyzer (Thermo Fisher Scientific Inc, model 49C, Franklin, MA) zeroed through an $O_3$ scrubber in the instrument, with LOD of 1.0 ppb and averaging time of 20 seconds. The analyzer was calibrated before and after each field deployment using an $O_3$ calibrator (Thermo Fisher Scientific Inc, model 49C, Franklin, MA). We note that $O_3$ may be overestimated due to interferences from VOCs emitted by the fire (Long et al., 2021), but the instrument used has been found to be in agreement with a federal reference method (Gao and Jaffe, 2017). $NO_x$ species were measured using a chemiluminescence $NO$-$NO_2$-$NO_x$

analyzer (Thermo Fisher Scientific Inc, model 42i, Franklin, MA). The $NO_x$ analyzer was calibrated automatically
every 6 hours, using NO and $NO_2$ calibration standards purchased from Airgas (Radnor, PA) and has an LOD of 0.40
ppb.
PM$_{2.5}$ mass concentration was determined with a Tapered Element Oscillating Microbalance (TEOM) series
1400a ambient particulate monitor (Thermo Fisher Scientific, Franklin, MA) with data recorded at an averaging time
of 60 seconds and typical detection limit of 5.58 µg m$^{-3}$ determined by 3 standard deviations of blank (filtered ambient
air) measurements. This data was subsequently averaged to time intervals of 20 and 60 minutes to mitigate noise,
especially when sampling under background conditions. The TEOM series 1400a developed originally by Rupprecht
& Patashnick is a US-EPA approved instrument for measuring the mass concentration of ambient PM$_{2.5}$ and PM$_{10}$ and
could be used for Federal Equivalent Method (FEM) regulatory measurements (Liu et al., 2017a; Patashnick and
Rupprecht, 1991). It is a gravimetric measurement that determines the mass accumulated on a microbalance over a
specified time interval at a monitored sample air flow rate. The sample air is preconditioned to a temperature of 50 °C
to remove liquid water interferences (Patashnick and Rupprecht, 1991), which may lead to the evaporation of highly
volatile PM$_{2.5}$ components, potentially underestimating the total mass concentration. Mass concentration over an
averaging period is calculated from the difference recorded between successive intervals. Due to random fluctuations
in the instrument operation when concentrations are low, this can lead to negative numbers, illustrated by the frequency
distribution of high time resolution data recorded by one TEOM shown in Fig. S1. When determining the average
background concentration, we include the negative mass concentrations since converting negative concentrations to
one half the LOD or ignoring them will produce an average that is biased high. In 2021, PM$_{10}$ TEOMs were also
deployed but this was found to be highly influenced by pollen, which can be high in the springtime, and so the
measurement was discontinued. Regional hourly PM$_{2.5}$ mass was reported at two Environmental Protection Division
(EPD) sites. In the following analysis we compare the PM$_{2.5}$ measured within the Fort to the EPD measurements at
the Columbus Airport and Phenix City South Girard (PCSG) school shown on the map in Fig. 1a. At Columbus
Airport, the Teledyne T640, which is based on broadline spectroscopy, is used, while the Met One BAM-1022 mass
monitor is used in Phenix City, utilizing a beta attenuation technique.
PM$_{2.5}$ black carbon (BC) mass concentration was measured by aethalometers. A range of multi and single
wavelength instruments were deployed. Two were seven wavelength instruments (Magee Scientific, model AE33 and
model AE31, Berkeley, CA) with detection ranges of 0.1–100 µg m$^{-3}$ and averaging times of 60 and 120 seconds
respectively, one 2-wavelength aethalometer (Magee Scientific, model AE22, Berkeley, CA) of 0.1 µg m$^{-3}$ detection
limit and 60 seconds averaging time, and two single wavelength particle soot absorption photometers (PSAPs)
(Radiance Research, Seattle, WA) of sensitivity > 0.1 µg m$^{-3}$ for 60 seconds averaging time. For the multiwavelength
aethalometers, BC was determined from the light absorption at 880 nm using the manufacturer's specified mass
absorption cross-section (MAC) of 7.77 m$^2$ g$^{-1}$, whereas for the single wavelength PSAPs, BC was determined from
the optical absorption coefficient at 565 nm assuming a specific mass absorption cross-section of 10 m$^2$ g$^{-1}$ following
the manufacturer's specifications. Two spot samplings of the model AE33 corrected for mass loading errors. This was
not done in the other instruments and so the data of the aethalometers (AE31 and AE22) were corrected for loading
interference using the method of Virkkula et al. (Virkkula et al., 2007). PSAPs measurements were not corrected due
to unavailability of scattering coefficients needed for correcting filter-based PSAP measurements (Bond et al., 1999;
Virkkula et al., 2005), which may lead to 10-20% underestimation of BC at sites where PSAPs were installed.
Brown carbon (BrC) was calculated from the 7-wavelength aethalometer measurements. BrC is largely
produced from biomass burning (Hecobian et al., 2010; Laskin et al., 2015; Yan et al., 2018; Fleming et al., 2020) and
in the following analysis used as a unique indicator of biomass burning smoke. While a small amount of BrC can be
produced from mobile sources and other sources of incomplete combustion, in the US, its predominant source is
biomass burning (Jo et al., 2016; Hecobian et al., 2010). We calculate the light absorption of BrC at 365 nm as a
marker for BrC levels. Using the aethalometer data, the absorption coefficient, which corresponds to (BC+BrC), was
inferred by multiplying mass concentration at each wavelength by the corresponding MAC value provided by the
manufacturer (Magee Scientific, Berkeley, CA). The absorption coefficient at 365 nm was determined by
extrapolating the linear regression of log absorption coefficient vs log wavelength since the lowest wavelength at
which the aethalometer operates is 370 nm. The slope of the linear relationship represents the negative of the
absorption Angstrom Exponent (AAE), a parameter used to study the optical properties of the aerosol. BrC at 365 nm
was then calculated by removing the estimated contribution of BC at 365 nm assuming that BrC does not absorb at
880 nm and that AAE of pure BC is 1. BrC absorption at shorter wavelengths is the difference of aethalometer-
measured total absorption and the extrapolated BC absorption (Lack and Langridge, 2013). All data of the light
absorption of BrC discussed in this work corresponds to the absorption calculated at 365 nm. Both $AAE_{total}$ and $AAE_{BrC}$
were calculated as the negative slopes of log absorption coefficient of total (BC+BrC) and BrC respectively, as a
function of log wavelengths. For $AAE_{total}$ the fit included wavelengths 370–880 nm (i.e., 370, 470, 520, 590, 660,
880), whereas for $AAE_{BrC}$ the wavelengths ranged from 370 to 660nm (i.e., 370, 470, 520, 590, 660).
In our analysis, we used meteorological and fuel moisture data from the Remote Automated Weather Stations
(RAWS) available online (https://raws.dri.edu/index.html). The closest RAWS weather station to all sites is named
Ft. Benning Georgia (Fig. 1a). In each trailer, all instruments were connected to a laptop computer with remote access
to reduce personnel time spent at the sites. Sites were generally visited every 1 to 2 weeks during which regular
instrument checks and maintenance were performed, such as restoring power, changing filters (for TEOMs and
PSAPs), measuring and recording flow rates and other instrument performance parameters.

### 2.4. Tools and analysis methods

### 2.4.1. Normalized Excess Mixing Ratios

To account for dilution of species of interest in a smoke plume, Normalized Excess Mixing Ratios (NEMRs)
are used. The NEMR is the ratio of enhancement of a studied species above the local background concentrations to
the enhancement of a long-lived component co-emitted from the biomass burning event. CO is often used as the
reference species, i.e., NEMR of species X is $\Delta X/\Delta Y$, where Y is CO measured in the same sample as X. To determine
the NEMR of X and the contribution of smoke to X from an identified burning region, the background concentration
of X (i.e., concentration if no smoke emissions) is subtracted from the measurement. In our study we used the average
of the measurements before and after the smoke event as the background since sampling was not performed upwind
of the fire. This method is supported by the observation from multiple sampling sites of spatially uniform background
concentrations and, in most cases, very low background concentrations relative to those recorded in the smoke.
However, there is more uncertainty when calculating $O_3$ NEMRs due to significant levels and diurnal changes in
background concentrations. NEMRs can also be determined from the slope of linear regressions. Here, we determine
NEMRs in each smoke event for $PM_{2.5}$ mass, BC, and BrC normalized by CO by first removing background
concentrations for data recorded during the event and then calculating the slope by linear regressions (i.e., the slope
of $PM_{2.5}$ mass concentration, BC concentration, or BrC absorption at 365 nm versus CO concentrations to determine
the respective NEMRs).

**2.4.2. Determining Smoke Sources and Plume Age**

To match specific fires to observed smoke at the monitoring sites, several methods were used. Data from the
Fire Information for Resource Management System (FIRMS) provided active fire data based on thermal anomalies.
These are based on measurements from the Moderate Resolution Imaging Spectroradiometer (MODIS), carried by
Aqua and Terra satellites, and the Visible Infrared Imaging Radiometer Suite (VIIRS), carried by the Suomi National
Polar-orbiting Partnership (Suomi NPP) and NOAA-20 satellites. FIRMS provides live and historical fire maps and
data that can be accessed online (https://firms.modaps.eosdis.nasa.gov/). This platform can be used to pinpoint specific
locations and obtain distances between points, which is useful for identifying possible fires where smoke was
transported to the sampling site and the time for smoke transport when combined with wind speed and direction data.
Although the FIRMS fire map is updated every 5 minutes, the polar orbiting satellites pass over the location only
twice per day meaning that some fires starting and ending between satellite observations are not detected (Schroeder
and Giglio, 2018; Giglio et al., 2021). Also, small or relatively cool fires may not be detected, especially when there
is significant cloud coverage, thick smoke, or a continuous, thick forest canopy, which can block satellite detection of
prescribed understory burns in forests. Cloud coverage data are available online
(https://worldview.earthdata.nasa.gov) and satellite data, including MODIS/VIIRS overpass times, the number of
active fire detections per pass, and FRP for all fires that impacted the monitoring sites, can be downloaded from the
abovementioned FIRMS website. Burn data provided by Fort Moore were used with the FIRMS data to minimize
limitations with each method for identifying sources of observed smoke. For each of the 64 smoke events studied in
the paper, burn data are added to the supplementary material (Table S1). Additionally, temperature, relative humidity,
and fuel moisture data used can be accessed online through RAWS USA Climate Archive
(https://raws.dri.edu/index.html) at the closest to all sites weather station named Ft. Benning, Georgia.
The Hybrid Single-Particle Lagrangian Integrated Trajectory (HYSPLIT) model (Stein et al., 2015) was used
to calculate back trajectories from monitoring sites. This trajectory analysis was based on meteorological data derived
from the Weather Research and Forecasting (WRF) model (Shamarock et al., 2019) enhanced with grid nudging and
observational nudging (Deng et al., 2009; Liu et al., 2005), using a 20-minute timestep. The WRF domain settings are
shown in Fig. S2. The winds used in the trajectory analysis are from the 1-km grid resolution domain. Each analysis
covered a total of 10 trajectories, all below the planetary boundary layer (PBL). HYSPLIT was run with 10-minute
timesteps, and the locations of fires were determined based on FIRMS data and the Fort Moore Fire Management
records.

## 3. Results and discussion

### 3.1. Assessment of PM$_{2.5}$ monitors and background concentrations

The focus of this analysis is on PM$_{2.5}$ mass concentrations from the prescribed fires. Aerosol particle mass concentrations measurements are difficult, especially at background conditions when concentrations are low. Calibrating instruments with known mass standards is also problematic. We performed intercomparisons between monitors including direct comparisons for two pairs (side-by-side) and intercomparison of background PM$_{2.5}$ mass concentrations measured by the study TEOMs to the values reported at state monitoring sites. For example, two TEOMs (used in main and T1293 trailers) collocated at Eglin Air Force Base in 2023 from March 19, 2023 at 8:00 till March 20, 2023 at 10:00, had an orthogonal regression slope of $0.98 \pm 0.09$, intercept of $0.45 \pm 0.37$ µg m$^{-3}$ and r$^2$ of 0.84 (see Fig. S3). The main trailer TEOM was also compared with the TEOM used on T1291 when they were collocated at the Georgia Institute of Technology from September 22, 2023 at 19:00 till October 07, 2023 at 14:00. Although measurement during that period was close to background levels, the comparison resulted in an orthogonal regression slope of $0.88 \pm 0.03$, intercept of $3.75 \pm 0.09$ µg m$^{-3}$ and an r$^2$ of 0.76 (see Fig. S4). The frequency distribution used to determine the mean values, and mean background values of the data recorded at the main trailer and the EPD sites in 2022 are shown in Fig. S5. The mean concentrations in 2022 were 7.02, 9.47, 9.01, 9.26, and 7.11 µg m$^{-3}$ at the main trailer, T1293, T1292, T1921, and T1290 respectively and 10.33 and 10.67 µg m$^{-3}$ at the Columbus Airport and PCSG school EPD sites respectively. Background air PM$_{2.5}$ mass concentrations were also determined by excluding smoke events (discussed below). The monthly backgrounds of PM$_{2.5}$ mass concentrations are shown in Table S2. Background concentrations were in the range of approximately 3–7 µg m$^{-3}$ for monitors at the Fort, and between 7 and 9 µg m$^{-3}$ at the state monitoring sites (Table S3). Higher background PM$_{2.5}$ mass concentrations at the state sites are likely due to the local anthropogenic (urban) influence. These comparisons provide confidence in the mass measurements that cannot be calibrated in a manner similar to gas monitors.

Background concentrations of CO and BC are also given in Table S2. Background CO ranged between ~ 150 and 200 ppb and background BC ranged between 0.14 and 0.57 µg m$^{-3}$. In terms of spatial variation within Fort Moore, background levels of measured species were slightly lower in sites located far from the main roads and training areas, such as measurements at the main trailer during May of 2021 and the entire 2022 season. No significant temporal variation is observed, although fires within and in the vicinity of the base increase during the transition from winter to spring, indicating that smoke was efficiently dispersed on time scales of approximately one day. Frequent smoke events where concentrations of the various measured species were substantially above these background levels were observed during the 2021 and 2022 field deployments.

### 3.2. Study of fires at Fort Moore during 2021 and 2022

### 3.2.1. Overview of the Smoke Detected

We first present an overview of the measurements at Fort Moore during two burning seasons. In the 2021 season, only one research (main) trailer was deployed. In the following year, four more were deployed for a total of five sites.

On March 18, 2021, a fully equipped trailer was deployed in the Northern boundary of Fort Moore, and we sampled at that location until April 26, 2021. It was then moved to the center of the Fort for sampling from April 26 to May 15, 2021 (see Fig. 1a). During this period, peaks of measured species were observed, as shown in the time series of $PM_{2.5}$ mass in Fig. 2b. A peak of a measured species is defined as the highest value observed within the data points, spanning from an initial rise until a return to background levels. Maximum $PM_{2.5}$ mass concentrations reached 2000 $\mu$g m$^{-3}$ for 20-minute averaged data and 1400 $\mu$g m$^{-3}$ for hourly-averaged data (Table S4). A total of 11 $PM_{2.5}$ peaks with mass concentrations greater than 35 $\mu$g m$^{-3}$ were recorded. In 2022, over the course of the entire burning season, 32 days recorded a total of 53 $PM_{2.5}$ mass concentration peaks greater than 35 $\mu$g m$^{-3}$ across the five measuring sites, as shown in Fig. 2c with similar high concentrations, reaching 841 $\mu$g m$^{-3}$ for 20-minute averaged and 513 $\mu$g m$^{-3}$ for hourly-averaged data (Tables S5 to S8).

We focus on smoke plumes with higher $PM_{2.5}$ mass concentrations to identify their sources and estimate the emissions and evolution of $PM_{2.5}$ mass because the burning areas are readily identified (e.g., detected remotely by satellite) and the plume can be easily delineated from the background. An increase in measured species is considered a peak, or event, when the 20-minute average $PM_{2.5}$ mass is greater than 35 $\mu$g m$^{-3}$ and the 40-minute average $PM_{2.5}$ mass concentration (average of two consecutive measurements) is larger than 30 $\mu$g m$^{-3}$. This excludes shorter transient events that includes a passing vehicle that can occur at measuring sites near training areas.

The large peaks in $PM_{2.5}$ mass are always accompanied by an increase in CO, BC, and BrC. Figure 3 shows the scatter plots of 20-minute averaged data collected in 2021 and 2022. The linear relation between $PM_{2.5}$ and CO, BC, and BrC during events resulted in an r$^2$ of 0.85, 0.68, and 0.71 respectively. On the other hand, for non-events data, which include all observations during the entirety of the measurement period, r$^2$ drops to 0.12, 0.33, and 0.17 for $PM_{2.5}$ mass vs CO, BC, and BrC respectively. These correlations suggest that the events identified correspond to periods of measuring smoke from biomass burning sources. However, it is important to note that variability still exists in slopes among different events, which will be explored and discussed in later sections.

### 3.3. Determining smoke sources

To study the emission and evolution of smoke plumes and make our measurements useful for evaluating smoke transport and dispersion models, we aim to link identified smoke plumes to specific burn areas and determine their transport time. Attribution of the smoke to specific fires is also useful for assessing the impacts of a specific prescribed burning program, such as the one at Fort Moore. Identifying the location of prescribed fires was complicated by several factors. In this study, we had limited beforehand information on the timing and location of planned burns from the burn managers. Moreover, smoke from other sources, such as prescribed and wildfires in the region, but not within the Fort, as well as uncertainty and variability in wind patterns at the time of burning, led us to utilize multiple methods to determine the source of each identified smoke episode.

Our analysis started by using satellite data from FIRMS to identify locations of fires (when the satellite passed overhead). After the end of the study, those locations were verified by cross-referencing with the Fort Moore fire management reports, which provided locations and acreage of prescribed burns and ongoing wildfires exclusively within the Fort for each day. Afterwards, we pinpointed the source of smoke that reached the monitors by averaging

the wind vectors at and before the peaks using the meteorological data from RAWS. This provided the expected
general upwind region the smoke likely came from. We also used the HYSPLIT model to conduct back trajectory
analysis from the measurement trailer for 8 hours prior to ascertain if the airmass containing the measured smoke had
passed the satellite-identified hot spot or the units reported as burnt by the Fort's Fire Management. HYSPLIT initial
altitudes were determined by the PBL height, where trajectories for 10 equally distributed altitudes between 10 m
above the surface and the top of the PBL were generated for each simulation. For example, if the PBL height was 100
m, trajectories were calculated at 10, 20, 30, 40, 50, 60, 70, 80, 90, and 100 m.
Through the systematic combination of these methods, we attempted to identify specific fire sources
associated with each observed smoke event and the time of transport of the smoke from the fire to the measurement
site (referred to as smoke age). This procedure was successful for 61 out of 64 of the identified smoke events. We
failed to identify 3 events that had no apparent source in agreement with the studied wind patterns. Moreover, of the
61 identified smoke events, 7 events were matched to different sources using the observed wind vector method versus
using the HYSPLIT trajectories, 7 events were matched to sources using HYSPLIT only, and 5 events were matched
to sources using wind vector method only.
The variability of smoke sources determined in some cases is attributed to the difference between wind
direction used by HYSPLIT and that recorded by RAWS used for the wind vector calculation. In HYSPLIT, wind
data are derived from the three-dimensional wind fields predicted by the application of the WRF model. Figure 4
shows a comparison between modeled and observed wind direction during the events identified in 2021 and 2022 at
the main trailer. A closer alignment in wind direction is observed during higher-speed wind conditions.
As an example of source determination, Figure 5a shows the time series of CO, $PM_{2.5}$ mass, BC, and BrC
during three smoke episodes recorded on April 6, 7, and 8, 2021, which are indicated by blue, yellow, and green
shading, respectively. Along the top of the graph are the hourly averaged wind vectors based on data from RAWS.
Note the high correlation between $PM_{2.5}$ mass and CO concentration and BrC absorption coefficient indicating that
the $PM_{2.5}$ peaks were due to smoke. During those three days, the three events were measured during late evening,
nighttime, and early morning periods. In each case, there is a time delay between when the burning occurred and when
the plume was measured, due to the transport time. In all three cases, burning regions at the Fort were identified as
the source. Consider the first smoke event detected at the trailer between 1:00 and 11:00 on April 6, 2021 (blue shaded
region in Fig. 5a). Figure 5b shows the map of the Fort and FIRMS satellite data on the day before (April 5, 2021)
indicating 2 hot spots on the base, which were later verified in the fire report as burning of 2 units and 4 sections of a
third unit. Both burns were to the south and south-southeast of the trailer, and the winds were from the westerly during
the daytime on April 5, 2021. By midnight, the wind direction shifted, with air flowing from the south and the
southeast, transporting smoke to the trailer's location, leading to elevated concentrations of species on monitors. Wind
speeds were very low at night. At about 8:00, wind speed increased, its direction changed, and concentrations of the
species all dropped.
Burning of other units took place on April 6, 2021 at distances 0.8, 2.1, 2.5, 6.3, and 7.2 miles from the trailer.
The level of measured smoke products started increasing in the evening after the winds became southwesterly and
stayed high until the morning of the next day (April 7, 2021) (yellow shaded region in Fig. 5a). Later at night of April
7 (green shaded region in Fig. 5a), concentration levels increased slightly after the burning of two connected units to
the south of the base during the daytime of April 7, 2021 at a distance ranging between 10.8 and 12.5 miles as indicated
by the Fort's Fire Management and seen on FIRMS. HYSPLIT back trajectory analysis, shown in Fig. 5 (e, f, and g),
was conducted for assessing our conclusion on the sources, especially in cases of wind variation and/or multiple fires
such as for the peaks monitored on April 6 (blue shaded region in Fig. 5a) and April 7, 2021 (yellow shaded region in
Fig. 5a). Since there are multiple fires on the Fort all in the same southern direction relative to the trailer, the exact
source cannot be determined solely based on wind vectors from RAWS data. In these cases, HYSPLIT back
trajectories help to pinpoint the exact fire or fires contributing to the smoke event observed. In both cases on April 6
(blue shaded region in Fig. 5a) and April 7, 2021 (yellow shaded region in Fig. 5a), the closer fire was the source of
smoke as shown in Fig. 5e and 5f.
**3.4. Determining smoke age**
An estimate of the smoke age is needed to separate fresh from aged smoke to estimate emissions of various
species (i.e., in fresh smoke) and the changes in their concentrations with plume age. The physical age of smoke is the
time it takes the smoke to be transported from the source to the monitoring sites. Following the concept presented for
source identification, the transport time of smoke is estimated by averaging wind speed over the period it takes for the
smoke to travel from the fire to the measurement sites, determined by iteration (mean wind speed recalculated with
new transport time, until convergence). When the average wind speed in the hour leading up to the peak does not
result in a smoke age of one hour or less, we begin iterative steps by calculating the average wind vector for additional
one hour increments at a time. A detailed example on using average wind vector in estimating the physical age of
smoke is provided in the Supplemental section S.1. It is important to note the uncertainty in the estimated smoke age
using this method for smoke monitored before and after the peak (maximum concentration), particularly when the
smoke event duration (from the start to the end of smoke monitoring) is prolonged, and when wind conditions are
highly variable. The age was also determined from the HYSPLIT back trajectories as the time when the lowest
trajectory intersects the source of smoke identified. The backward trajectory is initiated from the start time of the
smoke event. Due to uncertainties in the WRF simulated winds, particularly at night when wind speeds are low, the
backward trajectory occasionally missed the source. Therefore, a series of HYSPLIT simulations with 20-minute
intervals from the event start time until the source of smoke could be identified were conducted. The 20-minute interval
was chosen based on the temporal resolution of the WRF data.
For the three events discussed in Fig. 5, physical ages estimated using the wind vector averaged from
observed RAWS wind data are 75 minutes for April 6 (blue shading), 14 minutes for April 7 (yellow shading) and
162 minutes for April 8 (green shading), 2021 events. For the same events and using HYSPLIT trajectories closest to
the surface and passing through the identified sources, ages were estimated as 130 minutes for April 6 (blue shading),
10 minutes for April 7 (yellow shading), and 40 minutes for April 8 (green shading). Based on our analysis, April 6
(blue shading) stands out as the only case where the HYSPLIT age exceeds that estimated using the mean wind vector
for the same fire source. The difference between modeled and observed wind for these three instances was further
investigated by comparison with the observed wind at Columbus Airport. As shown in Fig. S6, the wind direction
observed at the airport aligns more closely with that observed at the RAWS site in Fort Moore (though with faster
winds at the airport, likely due to the forest canopy effect on wind flow) than with the WRF modeled winds at both
sites. However, it is difficult to determine which method is more reliable for studying any specific smoke event. For
all the smoke plumes identified, the age of smoke estimated based on HYSPLIT back trajectories ranged from 10
minutes (single timestep of trajectory) to 6 hours (36 timesteps), and from a few minutes to 8 hours based on average
wind vector method (Table S10). A comparison summary between wind speeds observed by the RAWS and those
modeled by WRF during all the events identified in 2021 and 2022 at the main trailer is shown in Fig. 6a. The observed
weak correlation ($r^2 = 0.29$) could be due to several factors. For the wind vector analysis, observed winds are measured
at one location and 2 meters above ground level with a single monitor, which may not accurately represent the wind
patterns along the entire smoke transport path, especially in forested areas where the canopy can affect the wind flow
(Mallia et al., 2020). On the other hand, WRF simulates winds for 34 layers at different altitudes from 10 m, being the
lowest, to levels higher than the PBL. HYSPLIT applies bilinear interpolation to the data from WRF for the 10
trajectories that it calculates, introducing additional uncertainty to the wind patterns used in the simulations. Although
the comparison between ages estimated based on the two different methods resulted in reasonable correlation ($r^2 =$
0.59), the slope clearly indicates a significantly higher estimation of age when using the wind vector method,
particularly for more aged smoke events, as shown in Fig. 6b, where ages from the two methods show stronger
agreement for fresh smoke. This can be attributed, in many cases, to the uncertainty in observed winds during low-
speed wind conditions, the measurement being far from where winds are observed (RAWS), and most importantly
that RAWS measures winds at 2 m above ground level whereas smoke transport happens at higher altitudes with
stronger winds. There are additional discrepancies resulting from wind variation at each altitude at which HYSPLIT
is running.

### 3.5. Limitations of the fixed site method

The goal of this project is to study the emissions and evolution of smoke from prescribed fires and provide
data to test model simulations and assessments of prescribed burning impacts. Some limitations and challenges are
associated with our approach of collecting data from a network of fixed sites.

### 3.5.1. Identification of burning regions

First, due to the limitations of satellite fire detection, some fires were not seen in FIRMS satellite detection
data but were subsequently identified from the fire management report, such as the prescribed fires on March 23,
2021, shown in Fig. S7a. The 20-minute averaged $PM_{2.5}$ mass concentration at the trailer increased to 74.8 µg m$^{-3}$, and
to 47.8 µg m$^{-3}$ hourly average at the EPD site located off-base at the Columbus Airport in the afternoon of March 23,
2021, as shown in the time series of Fig. S7b. This increase was accompanied by an elevation in the levels of CO,
$PM_{2.5}$ mass, BC, and BrC measured at the trailer. This is an example of burning on the Fort likely affecting the nearby
urban population. Prevailing winds were from the southeast at the time of the smoke event, as can be seen from the
wind vectors presented on the same time series in Fig. S7b. However, FIRMS satellite data showed no hotspots on the
Fort during the entire day. After checking the fire management report for 2021, prescribed burns for 3 units located in

the east central part of the Fort at distances ranging from 8.1 to 14.7 miles from the trailer were identified. Looking at either the wind vector at the time of the peak or the HYSPLIT back trajectories, the source of the smoke event identified on March 23, 2021 matches the closer prescribed burn conducted on the Fort.

Another issue with this approach is that relying only on data from the burning authorities at Fort Moore can, in some cases, be insufficient due to the lack of information about fires taking place off-base by landowners, such as the off-base fire seen on FIRMS during three overpasses of satellites at 12:38, 13:54, and 14:42 (Fig. S8a). On May 9, 2022 at 16:30, monitored species increased at the main trailer and 20 minutes average of $PM_{2.5}$ mass reached 52.3 $\mu g\ m^{-3}$ (Fig. S8b). The Fort's Fire Management reported no prescribed fires and one wildfire in the southern part of the base with an indication of zero probability of smoke from that fire reaching the trailer based on wind patterns. Based on both wind vectors and HYSPLIT simulations, the source of the event was identified as an off-base fire detected to the northeast of Fort Moore. The same smoke event was also observed at multiple trailers operating at the time and will be discussed more in the following section.

**3.5.2. Identifying a specific fire impacting the site when multiple burning is occurring**

When multiple fires are taking place simultaneously in varying wind conditions it can be difficult identifying the specific fire impacting the site, which can lead to uncertainty in the smoke age. This occurred in smoke detected around midnight on March 14, 2022 (see Fig. S9a). Relying on wind data, the smoke source is likely one or more of the fires on the east and/or southeast side of the base with a zero probability of it being one of the fires in the northern part of the base. HYSPLIT may help in narrowing down the possibilities of the smoke source (Fig. S9b), but there is still uncertainty in linking the specific fire to the observed event.

When several burning units are in close proximity and near the measurement site, identifying the specific source and smoke age can also be difficult (for example see Fig. S10). In this case burning in three units indicated by the Fort's Fire Management occurred at the same time close to each other and the trailer (distances of 0.6, 1.4, and 2.2 miles from the trailer). HYSPLIT trajectory at lowest altitude passes near (to the east), but not over the prescribed fires. Wind direction at the time of the event suggests influence of a minor portion from the northern part of the fire. It is important to note that in such cases, transport near the surface may be heavily influenced by fire-atmosphere interactions, making it difficult to rely on data from RAWS or WRF simulations as accurate indicators of atmospheric flows close to an active fire.

We note that there is no direct correlation between the amount of smoke reaching the trailer, i.e., measured species concentrations, and the distance of the fire from the monitoring site. The relation depends on the smoke transport and dispersion that may allow smoke to either directly hit the measuring site, partially reach the measuring site, or pass above the trailer with little or no smoke detection by the monitors. To illustrate this, we compare three case studies. Looking again at the smoke event of February 11, 2022 shown in Fig. S10, smoke reaching the trailer from 0.6 to 2.2 miles fires resulted in a 20-minute maximum $PM_{2.5}$ mass of 62.8 $\mu g\ m^{-3}$ and CO concentration of 1.3 ppm at 13:30. On February 12, 2022 (Fig. S11 a and b), smoke from burns of units at distances 4.3 to 4.6 miles from the trailer, caused an increase in 20-minute $PM_{2.5}$ mass concentration to 60 $\mu g\ m^{-3}$ and CO to 0.9 ppm at 13:50, whereas on the night of April 4, 2022 until the morning of April 5, 2022 (Fig. S11 c and d), smoke from fires 3.8 and 3.9 miles

from the trailer, caused an increase of 20-minute $PM_{2.5}$ mass concentration to 319 μg m$^{-3}$ and CO to 3.0 ppm at 1:10
over a longer smoke monitoring period. The much higher $PM_{2.5}$ mass concentrations measured on April 4, 2022
suggests that the trailer received a more direct smoke hit on that day than on February 11, 2022 or February 12, 2022,
despite the fire being closer on February 11 and having a very similar distance to the one detected on February 12.
This can also be attributed to the much lower nighttime PBL on April 4, which was 9.8 m and caused all HYSPLIT
trajectories to overlap as shown in Fig. 12d. Emissions from a smoldering fire with very little buoyant energy were
most likely trapped in this shallow layer leading to high concentration measurements. During the daytime on February
11 and 12, higher PBL of 1645 and 1305 m, respectively, favored more vertical dispersion of smoke.

### 3.5.3. Smoke not detected although regions of burning identified

On certain days, based on the wind data and the information presented in the fire management report, it
appears likely that smoke from the fires at the base should reach specific monitoring sites. However, during those
instances, such as the situation on February 15, 2022 shown in Fig. 7, no significant smoke peaks were detected. To
explain this outcome, two HYSPLIT forward trajectory simulations were run. The simulations show that if the fire
starts at 10:00, the smoke will not intercept the monitor, but if it starts at 11:00, the smoke at higher altitudes has a
slight chance of reaching the monitor. Overall, regardless of wind direction favoring smoke transport to monitors,
other factors like dispersion and smoke plume behavior, such as lofting, play a significant role in the transport process.

### 3.6. Using multiple monitoring sites to increase chances of measuring smoke and studying smoke evolution

There are distinct advantages of setting up multiple measuring sites and studying smoke over an extended
period. First, it helps capture more smoke events, as seen during the 2022 study in comparison with that in 2021 when
a single trailer was used. It minimizes issues with predicting downwind locations and is not affected by uncertainty in
planned burning locations and times. Second, it reduces the labor and time required for relocating a single trailer and
setting it up several times throughout a prescribed burning period where burning occurs over different regions. Third,
it provides high spatial resolution and occasionally smoke from the same fire is detected at several sites, which can be
useful in studying smoke chemical evolution with higher certainty than studies of multiple plumes of varying ages
measured on different days.
An example of the same fire detected at several sites is shown in Fig. 8. On May 9, 2022, each of T1291,
T1292, Main Trailer, and T1293 detected an off-base fire taking place approximately 11 miles to the north northeast
of the base, as shown in Fig. 8a. T1291, the closest trailer to the fire, measured $PM_{2.5}$ mass and CO peaks at 15:10.
The time series of species measured in the various trailers is shown in Fig. 8. Subsequent peaks in $PM_{2.5}$ mass, CO,
BC, and $O_3$ concentrations were recorded at T1292, at 15:50, then at the main trailer at 16:30, and finally at T1293,
the furthest trailer from the fire, at 18:10 local time. For $O_3$, note the $O_3$ enhancement ($\Delta O_3$) superimposed on the
diurnal $O_3$ trend. The ages of the smoke detected based on wind vector analysis were 266, 296, 330 and 480 minutes,
for the various trailers. The difference in smoke age is close to the difference in peak arrival times with maximum
$PM_{2.5}$ mass concentration observed at 15:02, 15:53, 16:25, and 18:16 on T1291, T1292, T1293, and T Main,
respectively. The differences in peak concentrations can be due to a number of factors, including changes in fire
emissions with time, extent of plume dilution with distance from the fire and changes in what portions of the plume
were measured due to changes in winds. Wind vectors are shown at the top of plots in Fig. 8. Wind direction and
speed varied during the period when the plumes were recorded; wind direction was between 52° and 86° from 11:00
till 14:00 and speeds between 3 and 7 mph on May 9, 2022. A shift in wind direction to 348° at a speed of 4 mph
happened at 15:00. Then, the wind direction fluctuated between 11° and 44°, before wind speed decreased to 0 mph
at 20:00 and remained calm until the morning of May 10, 2022. Normalizing these plume data by a stable smoke
tracer, such as CO, can account for some of these factors when comparing emissions and evolution of various plume
properties.

### 3.7. Interpretation of measurements to characterize smoke emissions and evolution

### 3.7.1. $PM_{2.5}$ emissions

We used the normalized excess mixing ratio (NEMR) to study the emissions of $PM_{2.5}$ species and their
evolution in the various measured smoke plumes. The NEMRs determined from the linear regression slopes of $PM_{2.5}$
species (mass concentration, BC concentration, BrC absorption versus CO, with backgrounds subtracted) and
correlation values ($r^2$) for all smoke events are summarized in Table 1. $PM_{2.5}$ mass concentration NEMRs from other
studies are summarized in Table S11.
The NEMR of fresh smoke near a fire is interpreted as an emission ratio (ER), assuming the smoke has
undergone limited chemical and/or physical changes. ERs based on NEMRs are widely used (Liu et al., 2017b; Collier
et al., 2016; Burling et al., 2011; Gkatzelis et al., 2024). They are compiled in reviews and emission inventories for
ambient (Andreae, 2019; Prichard et al., 2020) and laboratory fire studies (Yokelson et al., 2013), and for evaluating
or making model predictions (Xiu et al., 2022; Jaffe et al., 2022).
By focusing on fresh smoke (age less than 1 hour), the emissions ratios (ER) of the prescribed fires can be
estimated and compared to those from other studies. The $PM_{2.5}$ mass concentration ER ranged between 0.04 and 0.18
$\mu g \ m^{-3} \ ppb^{-1}$ and is shown in Fig. 9. These ERs are comparable to other prescribed fires measured at both ground level
(Alves et al., 2010; Desservettaz et al., 2017; Korontzi et al., 2003; Balachandran et al., 2013) and aloft in airborne
studies (Sinha et al., 2003; May et al., 2014; Gkatzelis et al., 2024; Travis et al., 2023) that span a large range of
burning conditions and fuels (details are provided in Table S11). The mean $PM_{2.5}$ mass concentration ER for our data
is $0.117 \pm 0.045 \ \mu g \ m^{-3} \ ppb^{-1}$ and that of these other prescribed fire studies are $0.098 \pm 0.034 \ \mu g \ m^{-3} \ ppb^{-1}$ for ground-
based and $0.188 \pm 0.154 \ \mu g \ m^{-3} \ ppb^{-1}$ for airborne measurements. There is substantial and similar variability in the
ground-based measurements of prescribed fire ERs in this study relative to other studies. More recent airborne-
measured prescribed fires have reported substantially higher ERs (Fig. 9). Smoke transported for 10 minutes from the
Blackwater river state forest prescribed fire reported by Gkatzelis et al. had an ER of $0.462 \ \mu g \ m^{-3} \ ppb^{-1}$ (Gkatzelis et
al., 2024) and Travis et al. reported a range of $0.188-0.433 \ \mu g \ m^{-3} \ ppb^{-1}$ for 22 prescribed fires studied and grouped
into 4 categories based on fuel type (Travis et al., 2023). Figure 9 also shows comparisons with wildfires reported in
other studies (Liu et al., 2017b; Collier et al., 2016; Palm et al., 2020; Gkatzelis et al., 2024). Wildfire $PM_{2.5}$ mass ERs
are significantly higher than ERs for prescribed fires in this work, with ER ranges between 0.04 and 0.43 $\mu g \ m^{-3} \ ppb^{-}$
$^1$ and a mean of $0.264 \pm 0.091 \ \mu g \ m^{-3} \ ppb^{-1}$ for wildfires, and the difference is statistically significant (two-tailed p

value is < 0.0001). Lower $PM_{2.5}$ mass ERs from smaller prescribed fires has been noted in other studies (Liu et al., 2017b) and supports utilizing prescribed burning as a land management tool to limit wildfires. However, differences in altitude at which the measurements were made may have some effect on ERs. Selimovic et al. (Selimovic et al., 2019) noted that the $PM_{2.5}$/CO in ground-level smoke was about half of that observed from aloft apparently due to reduction in aerosol mass from evaporation of semi-volatile aerosol particle components resulting from higher surface temperatures compared to aloft. Pagonis et al. also found airborne OA NEMRs to be a factor of 2 higher than ground-based NEMRs giving the same interpretation (Pagonis et al., 2023). When comparing ERs of prescribed fires in ground versus airborne studies of prescribed fires, shown in Fig. 9, the mean of airborne studies is a factor of ~ 1.9 higher than ground-based studies and the difference is statistically significant (p value is 0.025).

This analysis assumes no significant changes in $PM_{2.5}$ mass for smoke less than 1 hour old. We have seen that smoke detected in the afternoon can have enhanced $O_3$ concentrations, which may also lead to secondary aerosol formation. Smoke plumes with enhanced $O_3$ are identified in the ERs shown in Fig. 9 and indicate no bias within the range of ERs recorded, suggesting possible secondary aerosol formation within the first hour following emissions does not contribute to the ER variability. We also did not find evidence of ERs depending on time of day. No difference was seen between ERs for fires that started on the same day of measurement (i.e., all detected after 9:00 and before 17:00), and those detected at night, after 17:00, or early in the morning corresponding to fires that started the day before the measurement, but still were estimated to correspond to smoke less than one hour old.

We also determined the ERs for BC and BrC. BC ERs were in the range of 0.008–0.022 $\mu g\ m^{-3}\ ppb^{-1}$ with a mean value of 0.014 ± 0.004 $\mu g\ m^{-3}\ ppb^{-1}$, which are within the range of NEMRs reported in other studies; 0.006 $\mu g\ m^{-3}\ ppb^{-1}$ for prescribed burns in southern African savanna forests (Sinha et al., 2003), 0.020 $\mu g\ m^{-3}\ ppb^{-1}$ for rBC (refractory BC) for prescribed burns of California chaparral forests (Akagi et al., 2012), 0.022 $\mu g\ m^{-3}\ ppb^{-1}$ for chaparral forests (May et al., 2014), 0.006 $\mu g\ m^{-3}\ ppb^{-1}$ for fires in Montane ecosystems (May et al., 2014), 0.018 for coastal plain ecosystems in South Carolina (May et al., 2014), and 0.004 $\mu g\ m^{-3}\ ppb^{-1}$ for large wildfires over the western US measured during FIREX (Gkatzelis et al., 2024).

The BrC ERs of fresh smoke events ranged between 0.151 and 0.689 $Mm^{-1}\ ppb^{-1}$ with a mean ± standard deviation of 0.442 ± 0.157 $Mm^{-1}\ ppb^{-1}$. There is limited published data on BrC ERs and NEMRs from prescribed fires and the measurement techniques of BrC vary between studies. Liu et al. (Liu et al., 2016) reported aircraft measurements of BrC at 365 nm inferred from PSAP absorption coefficients measured at two wavelengths (470 and 532 nm) with an ER of 0.223 ± 0.053 $Mm^{-1}\ ppb^{-1}$ for fresh agricultural fires in the southeastern US, which is lower than our mean, but falls within the range of values we observed. For large wildfires measured over the western US, Zeng et al. (Zeng et al., 2022) found for Photoacoustic Spectroscopy (PAS) measurements of BrC at a wavelength of 405 nm, the ER was 0.131 ± 0.001 $Mm^{-1}\ ppbv^{-1}$ in plumes < 2 hours old. These values are in the range we recorded, but the BrC ERs for the prescribed fires of this study are more variable.

### 3.7.2. NEMRs of all smoke events and their change with smoke age

Here we assess the overall variability in NEMRs for $PM_{2.5}$ mass concentrations, BC mass concentrations, BrC absorption coefficients, and AAEs from all the smoke events (including ages less than 1 hour) and assess possible

trends with smoke plume age. In this analysis, the observed changes with age are a combination of variability in emissions and evolution of the aerosol since it is not a Lagrangian experiment, meaning that we are not continuously tracking a specific air mass containing smoke particles over time. $PM_{2.5}$ mass concentration NEMRs varied between 0.04 and 0.47 µg m$^{-3}$ ppb$^{-1}$ for all reported events with a mean ± standard deviation of 0.155 ± 0.076 µg m$^{-3}$ ppb$^{-1}$ (median is 0.138 µg m$^{-3}$ ppb$^{-1}$). BC NEMRs ranged between 0.005 and 0.024 µg m$^{-3}$ ppb$^{-1}$ with a mean value of 0.013 ± 0.005 µg m$^{-3}$ ppb$^{-1}$. BrC NEMRs ($\Delta BrC/\Delta CO$) varied between 0.133 to 1.550 Mm$^{-1}$ ppb$^{-1}$. (Note that data collected on April 21, 2022 at trailer 1293 is an outlier with exceptionally high ERs for $PM_{2.5}$ mass concentration and BrC absorption coefficient. The ER for BC mass concentration, while elevated, falls within the observed range. This event corresponds to smoke from an identified prescribed fire at the Fort and has a relatively low $\Delta CO$ of 66.1 ppb, which is unexpected given the burn's proximity and the wind speed on that day, causing ERs to be significantly higher. The HYSPLIT back trajectory from the measuring site does not intersect with the fire but passes close to it. Although the FRP reported on FIRMS does not differ from that of other fires, and there is no significant difference in vegetation type or fuel moisture, the most likely explanation for this event is that the smoke passing through the measurement site was not a direct hit but from the diluted boundary of the plume, which may have undergone photochemical processing, leading to higher $PM_{2.5}$, BrC, and $O_3$ NEMRs.). The NEMRs are given in Table 1 for all smoke events data and plotted in Fig. 10 as a function of estimated smoke age determined from the wind vector and HYSPLIT analysis. From these plots we assess if there is any systematic evolution of the $PM_{2.5}$ mass, BC and BrC.

*Changes in PM$_{2.5}$ Mass Concentration NEMR with smoke age*: From Fig. 10a, $PM_{2.5}$ mass concentration NEMR shows substantial variability at all ages with no significant statistical difference or clear trend, however, NEMR tends to be lower for fresh smoke events (≤ 1 hour old) versus more aged plumes, possibly from secondary aerosol formation. Considering only smoke plumes in which $O_3$ enhancements were observed (i.e., smoke measured between 12:00 and 18:00), $PM_{2.5}$ mass concentration NEMR consistently increases with physical age ($r^2 = 0.65$), possibly evidence of secondary aerosol formation driven by photochemistry.

A range of results for changes in $PM_{2.5}$ mass concentration NEMRs in wildland fires have been observed in other studies, including systematic increases, little change, or decreases with smoke age. To the best of our knowledge, no ground-based studies have been conducted on the evolution of smoke from prescribed fires, but frequent airborne studies have investigated prescribed and wildland smoke aging because of the ability to spatially characterize a single plume. While studying two prescribed fires in SC, May et al. (May et al., 2015) observed no statistically significant net change in OA NEMRs near the source and downwind for smoke transported for ≤ 1.5 hours. One of the two fires was studied for longer, and results showed downwind OA NEMRs over 2 to 5 hours of transport significantly lower than the NEMRs at the source, suggesting a net loss of emitted OA. For wildfires, Collier et al. (Collier et al., 2016) found increases, little change, and decreases with smoke age in different wildfire plumes measured in Oregon. For the selected large wildfires in the western US in summer, Palm et al. (Palm et al., 2020) reported that the OA NEMR remained almost constant at a value of ~ 0.25 µg m$^{-3}$ ppb$^{-1}$ as the plume aged from 20-50 minutes to 6 hours. In their analysis of the data of wildland fires studied during FIREX-AQ campaign in 2019, Pagonis et al. report OA NEMR increased from 0.2 g g$^{-1}$ to 0.3 g g$^{-1}$ in 3 hours (Pagonis et al., 2023). While Garofalo et al. found no significant change of NEMRs between 0.5-8 hours transport of smoke from 20 western wildfires, they concluded that there was secondary

OA formation through oxidation driven condensation, but it was balanced by dilution-driven evaporation (Garofalo et
al., 2019). Gkatzelis et al. reported the NEMRs of some plumes that were more than an hour old and are shown in
Table S11 with their corresponding physical age (Gkatzelis et al., 2024). For the same fire (William's flat), the NEMR
was 0.331 µg m$^{-3}$ ppb$^{-1}$ at a physical age of 15 minutes that increased to 0.524 µg m$^{-3}$ ppb$^{-1}$ at 102 minutes (Gkatzelis
et al., 2024). Similar increase for the Castle fire was seen where the NEMRs reported are 0.204, 0.244, and 0.463 µg
m$^{-3}$ ppb$^{-1}$ at 25, 27, and 153 minutes respectively. For another fire (Horsefly), the NEMR was 0.398 µg m$^{-3}$ ppb$^{-1}$ at a
physical age of 65 minutes and remained at a similar value of 0.391 µg m$^{-3}$ ppb$^{-1}$ at 104 minutes. On average, the mean
NEMRs for plumes of physical age less than one hour, reported in their study, was $0.218 \pm 0.110$. This value is lower
than that of plumes older than one hour, which have a mean value of $0.391 \pm 0.131$ (Gkatzelis et al., 2024). Overall,
we find no trends in our data when considering all the smoke plumes detected, but for periods of expected
photochemical activity we observe consistent evidence for aerosol formation with plume age, which might be
attributed to the optically thin smoke that allows photochemistry throughout the plume compared to large optically
thick wildfires that leads to more complex photochemistry within the plume (Decker et al., 2021a).

651         We examined other factors that may contribute to variability of $PM_{2.5}$ mass NEMRs. No significant difference

was observed between on-base and off-base sources of smoke. Mean $PM_{2.5}$ mass NEMR of smoke originating from
outside the base is 0.208 (range 0.112–0.277 µg m$^{-3}$ ppb$^{-1}$), compared to 0.147 µg m$^{-3}$ ppb$^{-1}$ (range 0.042– 0.466 µg
m$^{-3}$ ppb$^{-1}$) for on base burning, which is not statistically different (two tailed p-value is 0.076). A preliminary
assessment using Google Earth satellite imagery and Landscape Fire and Resources Management Planning Tool
(LANDFIRE, https://www.landfire.gov/) does not show any visible differences in vegetation between the forested
areas burnt on and off the base. Additionally, no further information regarding the fuel types in the off-base lands
could be obtained. Just like no detected differences being observed between day/night $PM_{2.5}$ mass concentration ERs,
there was no significant difference (p-value is 0.169) between smoke plumes of all ages measured during the day
corresponding to fires occurring within a few hours from starting the burn (after 9:00 and before 17:00) (mean NEMR
= 0.178 µg m$^{-3}$ ppb$^{-1}$) and those monitored  at night and early in the morning corresponding to fires starting the day
before (after 17:00) (mean NEMR = 0.137 µg m$^{-3}$ ppb$^{-1}$), in contrast to an observed trend of $PM_{2.5}$ mass NEMR with
age for smoke with $O_3$ enhancement. This may suggest little night-time secondary aerosol formation (Brown et al.,
2013), but a more focused analysis is needed to better assess possible evidence for secondary aerosol formation. No
correlation was observed between $PM_{2.5}$ mass NEMRs and relative humidity ($r^2 = 0.08$) or fuel moisture data ($r^2 =$
0.04) for the smoke events in this study (Fig. S13). A weak positive correlation between air temperature and $PM_{2.5}$
mass NEMRs was observed, with an $r^2$ of 0.14 for all smoke events and $r^2$ of 0.44 for fresh smoke events. Many factors
could cause variability in $PM_{2.5}$ mass NEMRs, but no single factor could be identified when all data from this study
is grouped together.

670         ***Changes in BC and BrC NEMR with smoke age:*** BC and BrC NEMRs versus age are shown in Fig. 10b

and 10c with periods of $O_3$ enhancements identified. No trend in BC NEMRs with age is observed, as expected, since
BC is primarily emitted and largely nonvolatile. Lack of a trend supports this analysis approach, and all the BC
measured in events largely reflects BC variability in emissions relative to CO. BrC NEMRs are also highly variable
and have no trend with age for all the data or just the periods of $O_3$ enhancements. Since BrC can be both primary and

secondary, is semi-volatile, and undergoes photo-bleaching, a range of results on BrC evolution has been observed in past studies (Zhong and Jang, 2014; Saleh et al., 2013; Liu et al., 2016). Like BC, a similar large variability, with no trend, in BrC NEMRs with ages up to 8 hours has been observed for wildfires in the western US (Zeng et al., 2022; Sullivan et al., 2022; Palm et al., 2020) whereas in some cases consistent loss (bleaching) of BrC has been reported (Forrister et al., 2015). Optical properties of absorptive aerosol spectral properties characterized by AAE are shown in Fig. 11 as a function of age. Total absorption AAE values from the two trailers with 7-wavelength aethalometers (i.e., BC+BrC measured by the aethalometer) varied between 1.31 and 3.32 (mean ± stdev of 1.89 ± 0.23) and between 3.19 and 7.43 (mean = 5.00 ± 0.89) for BrC only. AAEs have no trend with age for either fresh smoke plumes or periods of $O_3$ enhancement. While our total AAE values are similar (Zeng et al., 2022; Strand et al., 2016; Marsavin et al., 2023) or sometimes lower (Liu et al., 2016; Forrister et al., 2015) than those in other biomass burning studies, it is indicative of the presence of BrC in the smoke plumes studied. As for BrC AAEs, our reported values are significantly higher than those reported for western wildfires, where BrC determined from the PAS had an AAE of 2.07 ± 1.01 (Zeng et al., 2022), indicating difference in BrC optical properties or with instrumentation, which needs further investigation. Selimovic et al. show that duff has the highest AAE of 7.13 (calculated from absorption data at 401 and 870 nm) when burnt, and it is typically consumed more in wildfires than in prescribed fires. However, the variability in optical properties is influenced more by the differential consumption of individual components than by the dominant tree species in the ecosystem (Selimovic et al., 2019).

## 4. Conclusion

We describe a ground-based observational study for characterizing smoke from prescribed fires based on continuous monitoring at multiple sites for an extended period in a regularly burned region. We focus on burning within a large military Fort in the southeastern US and identify the sources of the smoke to determine if it was within or outside the Fort and study emissions and evolution of smoke species. The method was successful in capturing a significant number of smoke events (64) monitored on 42 days and linked to 45 fires across 2 burning seasons. Source and age for each smoke plume detected was estimated. This allowed us to match 95 % of the identified events to their corresponding source and to calculate the estimated transport time of smoke from source to monitors. These data were used to characterize emissions and evolution of key smoke parameters through calculation of normalized excess mixing ratios (NEMRs), with CO as the conserved co-emitted species. Overall, $PM_{2.5}$ mass concentration NEMRs ($\Delta PM_{2.5}$ mass/$\Delta CO$) ranged between 0.04 and 0.47 µg m$^{-3}$ ppb$^{-1}$ with a study mean of 0.155 ± 0.076 µg m$^{-3}$ ppb$^{-1}$ (median is 0.138 µg m$^{-3}$ ppb$^{-1}$). For plumes less than 1 hour old the $PM_{2.5}$ mass concentration NEMRs were interpreted as a characteristic of the fire's emissions. Emissions ratios for fires of this study ranged between 0.042 and 0.176 µg m$^{-3}$ ppb$^{-1}$ with a mean of 0.117 ± 0.045 µg m$^{-3}$ ppb$^{-1}$ (median is 0.121 µg m$^{-3}$ ppb$^{-1}$). These emissions estimates are in the range reported in other ground-based studies for a range of fires and fuels but are lower than what has been reported for wildfire smoke measured from aircraft at higher altitudes. BC and BrC NEMRs and emission ratios are also reported. An analysis of $PM_{2.5}$ mass and BrC NEMRs changes with smoke age showed no consistent trends for all combined smoke plumes. However, $PM_{2.5}$ mass NEMRs did increase with age for smoke detected in the afternoon in plumes where $O_3$ enhancements were observed, indicating the formation of $O_3$ and secondary aerosol. This was not

observed for BrC NEMRs.  This data set will be used to assess models predicting the impact of prescribed fires on air
quality to enhance the use of prescribed burning in land management practices by minimizing impacts on populations.
**5. Competing interests**
The contact author has declared that none of the authors has any competing interests.
**6. Acknowledgements**
We thank Fort Moore authorities for hosting the field study, and to the members of the Natural Resources Management
Branch for sharing information about the burns. REA, DJT, GH and RJW were supported by the United States Army
Corps of Engineers under contract W912HQ-20-C-0019. ZL, YH, and MTO were supported by the Strategic
Environmental Research and Development Program (SERDP) through project RC20-1047.
**Author contribution:** REA and RJW wrote the paper. RJW, LGH, DJT, and MTO designed the experiment. REA
and DJT collected the data. REA, ZL, DJT, and RJW analyzed data. REA and ZL worked on the HYSPLIT analysis.
All authors reviewed and provided comments for the paper.
**Data availability:** Data are available in a publicly accessible repository: https://doi.org/10.5281/zenodo.11222295.

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

## Tables

**Table 1**. The PM$_{2.5}$ mass, BC, and BrC NEMRs relative to CO (based on regression slopes) and coefficients of determination ($r^2$) in the column to the right of each NEMR for the smoke events identified in this study*.

| Smoke event Date/Trailer | NEMR PM$_{2.5}$ mass (µg m$^{-3}$ ppb$^{-1}$) | $r^2$ | NEMR PM$_{2.5}$ BC (µg m$^{-3}$ ppb$^{-1}$) | $r^2$ | NEMR PM$_{2.5}$ BrC (µg m$^{-3}$ Mm$^{-1}$) | $r^2$ | Age estimated by wind vector (min) | Age estimated by HYSPLIT (min) |
|---|---|---|---|---|---|---|---|---|
| 3/23/21 T Main | 0.125 | 0.66 | 0.010 | 0.74 | 0.257 | 0.59 | 108 | 40 |
| 4/06/21 T Main | 0.097 | 0.90 | 0.012 | 0.96 | 0.187 | 0.82 | 75 | 130 |
| 4/07/21 T Main | 0.160 | 0.90 | 0.012 | 0.93 | 0.367 | 0.86 | 14 | 10 |
| 4/08/21 T Main | 0.105 | 0.90 | 0.005 | 0.84 | 0.199 | 0.85 | 162 | 40 |
| 4/14/21 T Main | 0.146 | 0.72 | 0.015 | 0.76 | 0.324 | 0.61 | 44 | 20 |
| 4/20/21 T Main | 0.080 | 0.74 | 0.011 | 0.83 | 0.151 | 0.63 | 5 | 10 |
| 4/21/21 T Main | 0.107 | 0.75 | 0.009 | 0.90 | 0.133 | 0.70 | 330 | 190 |
| 4/30/21 T Main | 0.141 | 0.94 | 0.007 | 0.95 | 0.319 | 0.87 | - | - |
| 2/11/22 T Main | 0.054 | 0.93 | 0.022 | 0.95 | 0.567 | 0.95 | 8 | 10 |
| 2/12/22 T Main | 0.066 | 0.82 | 0.018 | 0.96 | 0.514 | 0.93 | 60 | 50 |
| 2/13/22 T Main | 0.053 | 0.81 | 0.016 | 0.83 | 0.613 | 0.85 | 26 | 20 |
| 2/13/22 T Main | 0.042 | 0.86 | 0.014 | 0.89 | 0.689 | 0.85 | 30 | 20 |
| 2/26/22 T Main | 0.207 | 0.88 | 0.018 | 0.98 | 0.690 | 0.97 | 130 | 110 |
| 2/27/22 T Main | 0.119 | 0.70 | 0.010 | 0.87 | 0.334 | 0.91 | - | - |
| 3/01/22 T Main | 0.166 | 0.81 | 0.016 | 0.91 | 0.586 | 0.94 | 92 | 270 |
| 3/02/22 T Main | 0.129 | 0.75 | 0.020 | 0.87 | 0.608 | 0.87 | 60 | 40 |
| 3/04/22 T Main | 0.209 | 0.69 | 0.005 | 0.53 | 0.167 | 0.92 | - | 160 |
| 3/04/22 T Main | 0.121 | 0.89 | 0.012 | 0.98 | 0.454 | 0.97 | - | 40 |
| 3/07/22 T Main | 0.122 | 0.82 | 0.009 | 0.96 | 0.405 | 0.96 | 224 | - |
| 3/07/22 T Main | 0.170 | 0.66 | 0.012 | 0.97 | 0.338 | 0.89 | - | 10 |
| 3/14/22 T Main | 0.138 | 0.82 | 0.010 | 0.93 | 0.575 | 0.88 | - | 20 |
| 3/25/22 T Main | 0.090 | 0.78 | 0.009 | 0.86 | 0.375 | 0.91 | 5 | 10 |
| 3/29/22 T Main | 0.121 | 0.68 | 0.008 | 0.68 | 0.420 | 0.76 | 5 | 10 |
| 4/04/22 T Main | 0.129 | 0.90 | 0.009 | 0.96 | 0.551 | 0.92 | 168 | 130 |
| 4/25/22 T Main | 0.283 | 0.83 | 0.022 | 0.91 | 1.382 | 0.77 | 169 | 90 |
| 5/09/22 T Main | 0.237 | 0.96 | 0.008 | 0.94 | 0.324 | 0.94 | 330 | 150 |
| 3/21/22 T 1293 | 0.188 | 0.98 | - | - | - | - | 89 | 20 |
| 3/25/22 T 1293 | 0.158 | 0.93 | - | - | - | - | 45 | 30 |
| 3/26/22 T 1293 | 0.148 | 0.97 | - | - | - | - | 5 | 10 |
| 3/27/22 T 1293 | 0.176 | 0.84 | - | - | - | - | 5 | 10 |
| 3/28/22 T 1293 | 0.129 | 0.81 | - | - | - | - | - | 60 |
| 3/29/22 T 1293 | 0.093 | 0.87 | - | - | - | - | - | 210 |
| 4/05/22 T 1293 | 0.277 | 0.91 | 0.016 | 0.78 | 0.280 | 0.47 | - | 360 |
| 4/21/22 T 1293 | 0.466 | 0.98 | 0.024 | 0.83 | 1.55 | 0.48 | 78 | - |
| 4/23/22 T 1293 | 0.121 | 0.59 | 0.013 | 0.80 | 0.317 | 0.33 | 28 | 10 |
| 4/23/22 T 1293 | 0.165 | 0.97 | 0.014 | 0.96 | 0.354 | 0.94 | 48 | 10 |
| 4/24/22 T 1293 | 0.248 | 0.90 | - | - | - | - | 63 | 40 |
| 4/26/22 T 1293 | 0.182 | 0.96 | - | - | - | - | 106 | - |
| 5/09/22 T 1293 | 0.238 | 0.99 | 0.012 | 0.98 | 0.321 | 0.94 | 480 | 210 |
| 5/10/22 T 1293 | 0.112 | 0.92 | 0.008 | 0.83 | 0.406 | 0.78 | 474 | 160 |
| 5/11/22 T 1293 | 0.168 | 0.77 | - | - | - | - | 5 | 10 |
| 5/12/22 T 1293 | 0.119 | 0.94 | - | - | - | - | 5 | 10 |
| 5/09/22 T 1291 | 0.265 | 0.98 | - | - | - | - | 296 | 160 |

* The table lists all events where both PM$_{2.5}$ mass and CO concentration were both available. In some cases BC and BrC data was not available and left as blank values ( - ).

**Figures**

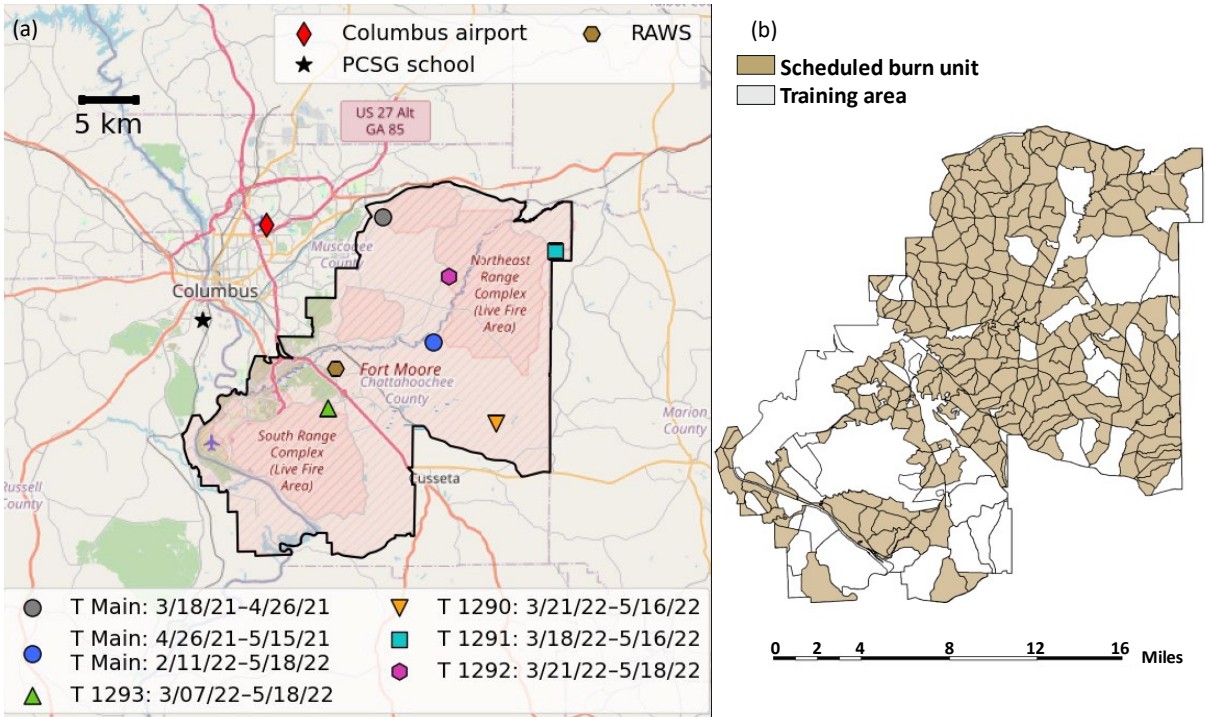


**Figure 1.** Study region overview. (a) Fort Moore map with the locations of trailers, RAWS weather station, and two state-operated
sampling sites, Columbus Airport and Phenix City South Girard (PCSG) school, are shown along with the location of the city of
Columbus GA. (b) Fort Moore map showing the planned burn units for the year 2021, sourced from Fort Moore authorities and
natural resources management team, with prevailing winds in the region.



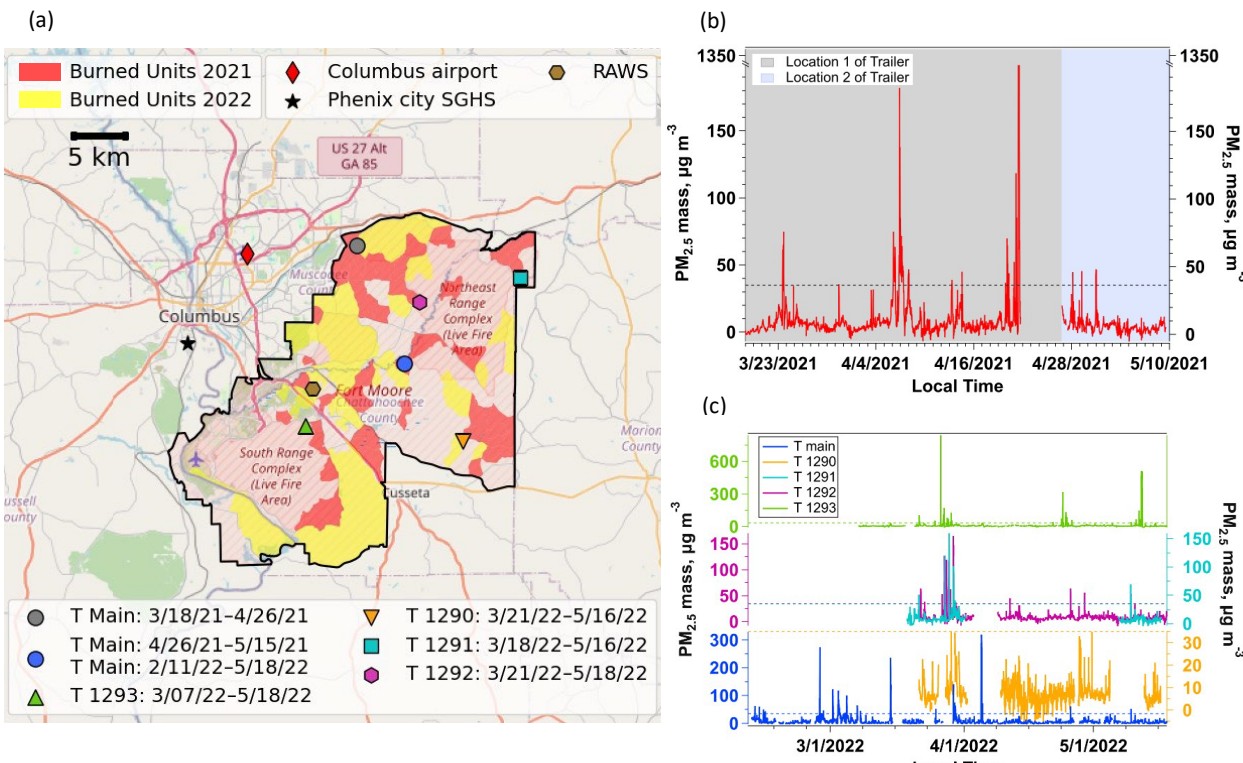


**Figure 2.** PM$_{2.5}$ mass measurements over two burning years. (a) Map of the burnt areas in the years 2021 and 2022 and locations
of monitoring sites. (b) Time series of 20-minutes average PM$_{2.5}$ mass concentration measured at the main trailer during the burning
season of 2021, and (c) 2022 across different sites. Dotted lines represent PM$_{2.5}$ mass concentration of 35 µg m$^{-3}$ above which peaks
were selected for detailed analysis.



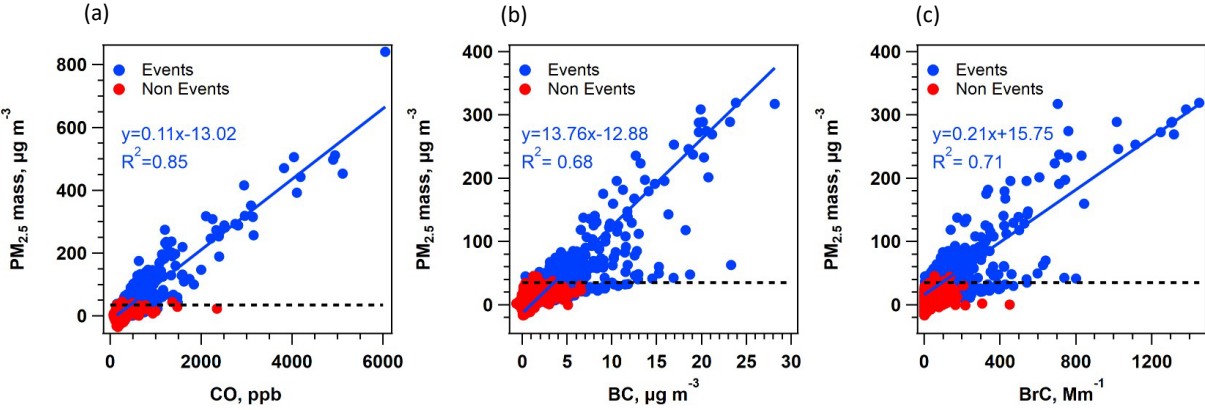


**Figure 3.** Correlations between PM$_{2.5}$ mass concentration and CO, PM$_{2.5}$ BC, and PM$_{2.5}$ BrC for measurements from the main trailer
in 2021 and 2022 and T1291 and T1293 in 2022. Blue data points are characterized as PM$_{2.5}$ events when the concentration is > 35
µg m$^{-3}$ averaged over a 20-minute period. In the plot all data associated with an identified event is shown as blue (This includes
event data down to the background levels before and after the peak). All other data (non-events) are shown in red. Slope is from
orthogonal distance regression (ODR) of the 20-minute averaged data during events periods.


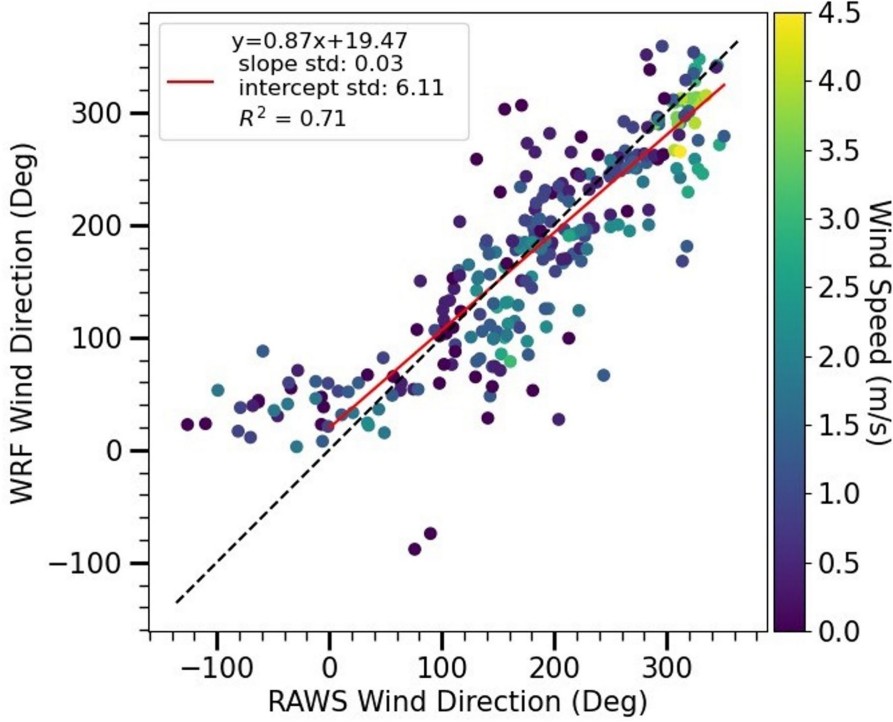


**Figure 4.** Comparison between wind direction modeled via WRF versus that recorded by the RAWS located at Fort Moore. Slope is from orthogonal distance regression (ODR).



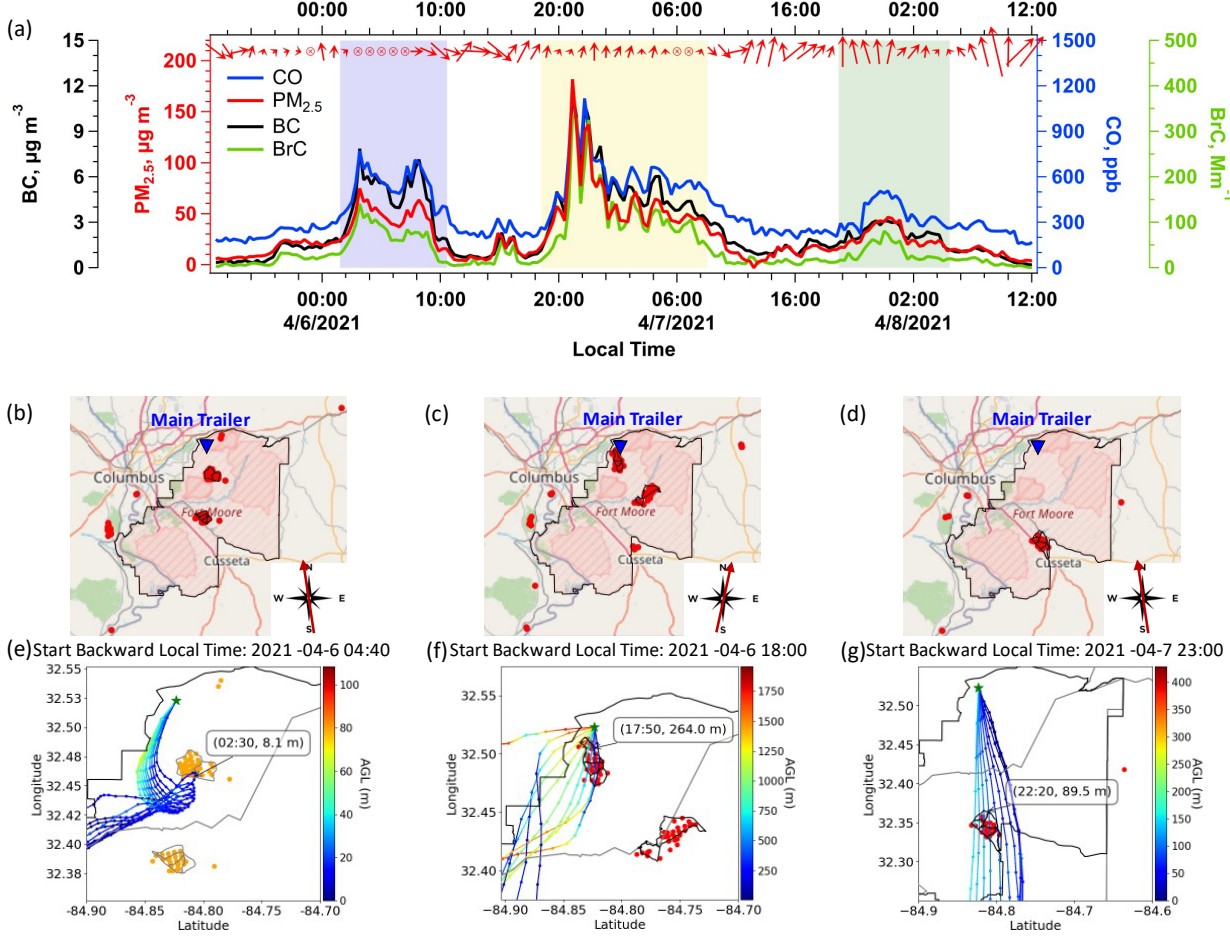


**Figure 5.** Three case studies illustrating the application of our method in determining the source of smoke events. (a) Time series
of species measured at the main trailer. Time resolution is 20 minutes for CO, PM₂.₅ mass, BC, and BrC. The wind vectors depict
hourly data obtained from RAWS, with the direction of the arrow indicating wind direction, and the length of the arrow representing
wind speed. (b,c,d) Maps of the Fort showing historical satellite data from the FIRMS website observed for April 5, 6, and 7, 2022.
Red dots represent fires detected by the satellite. (e,f,g) Are HYSPLIT back trajectories during the occurrence of each of the three
peaks. Date and time of the backward trajectory is indicated on top of each map. Time and height at which the trajectory crosses
the trailer is shown in the box inside each map. Red dots are fires detected by FIRMS the same day of the backward trajectory.
Orange dots are fires detected by FIRMS one day before the day of the backward trajectory. The colors of the traces in the back
trajectories indicate the height above ground level. Green star marks the location of the main trailer. Satellite overpasses times are
shown in Table S9.


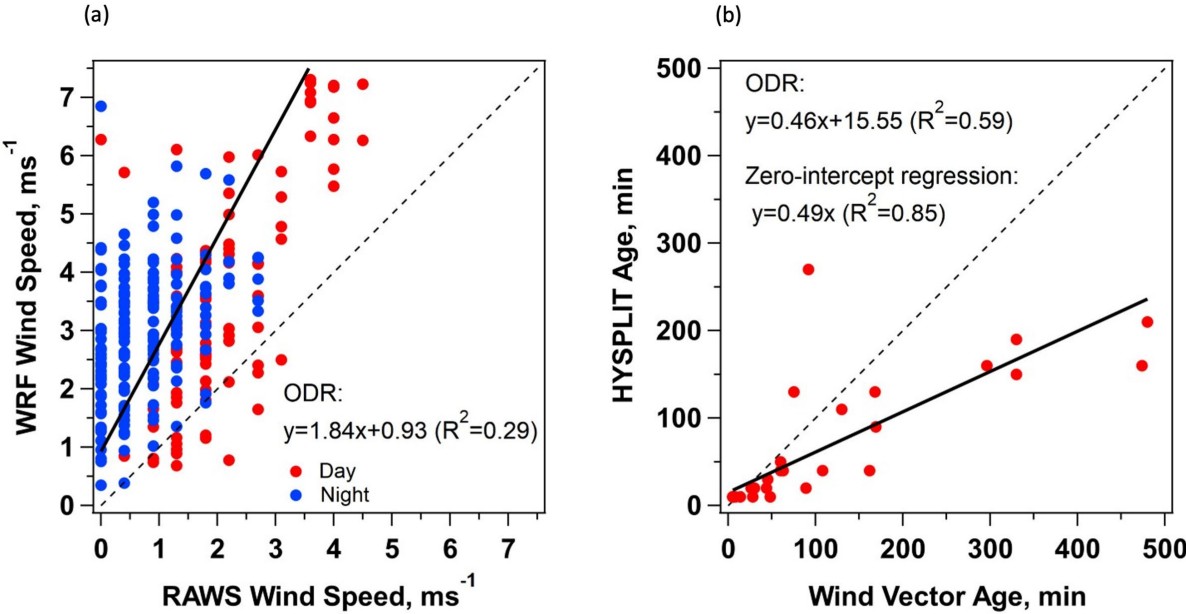


**Figure 6.** (a) Comparison between wind speed modeled via WRF versus that observed by RAWS located on Fort Moore. (b)
Comparison between age estimated using HYSPLIT model versus the wind vector method.


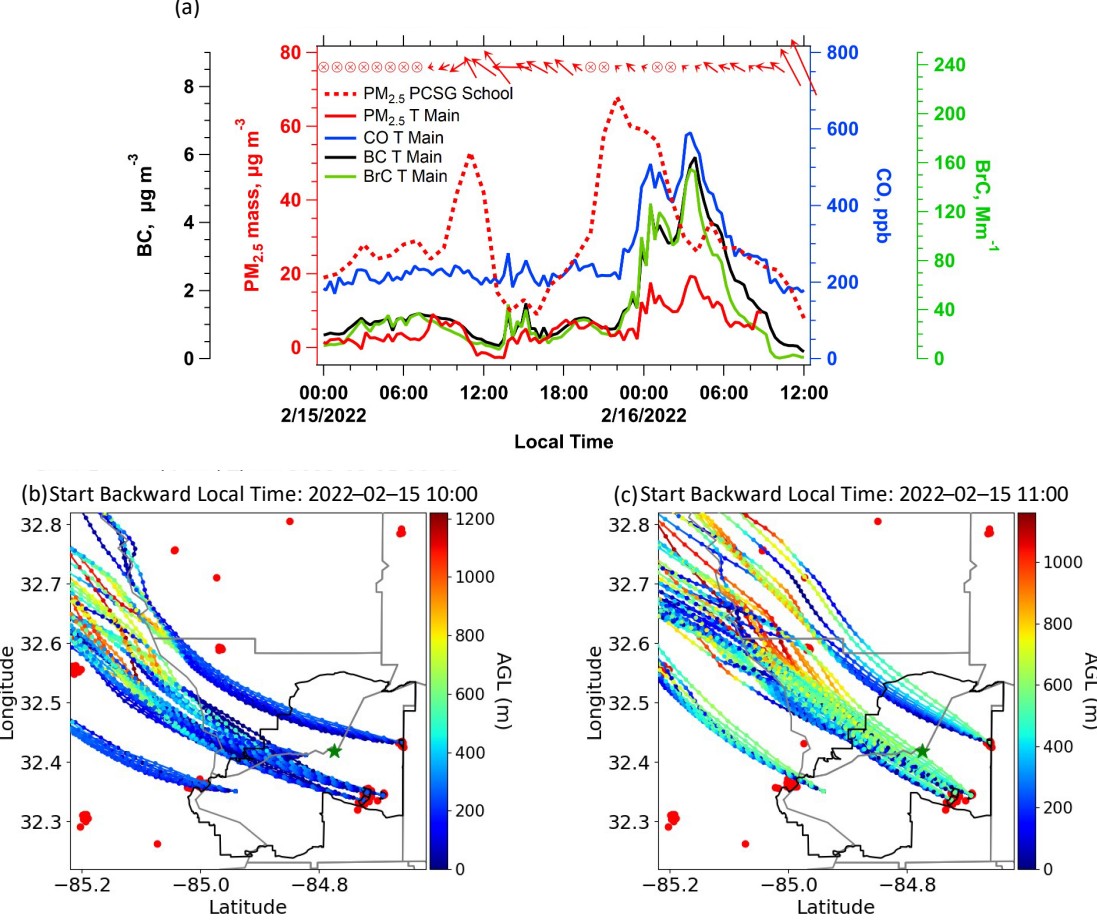


**Figure 7.** Case study of missing smoke at monitoring site despite expectations according to wind direction. (a) Time series of species measured at the main trailer. Time resolution is 20 minutes for CO, PM$_{2.5}$ mass, BC, and BrC. The wind vectors depict hourly data sourced from RAWS, with the direction of the arrow indicating wind direction, and the length of the arrow representing wind speed. Data from PCSG school are hourly averages; (b, c) HYSPLIT forward trajectories starting from the two prescribed fires on the base on February 15, 2022 at 10:00 and 11:00, respectively. Red dots are fires detected on FIRMS the same day (satellite overpass happened on February 15, 2022 at 12:54, 13:49, 14:32, and 14:36).


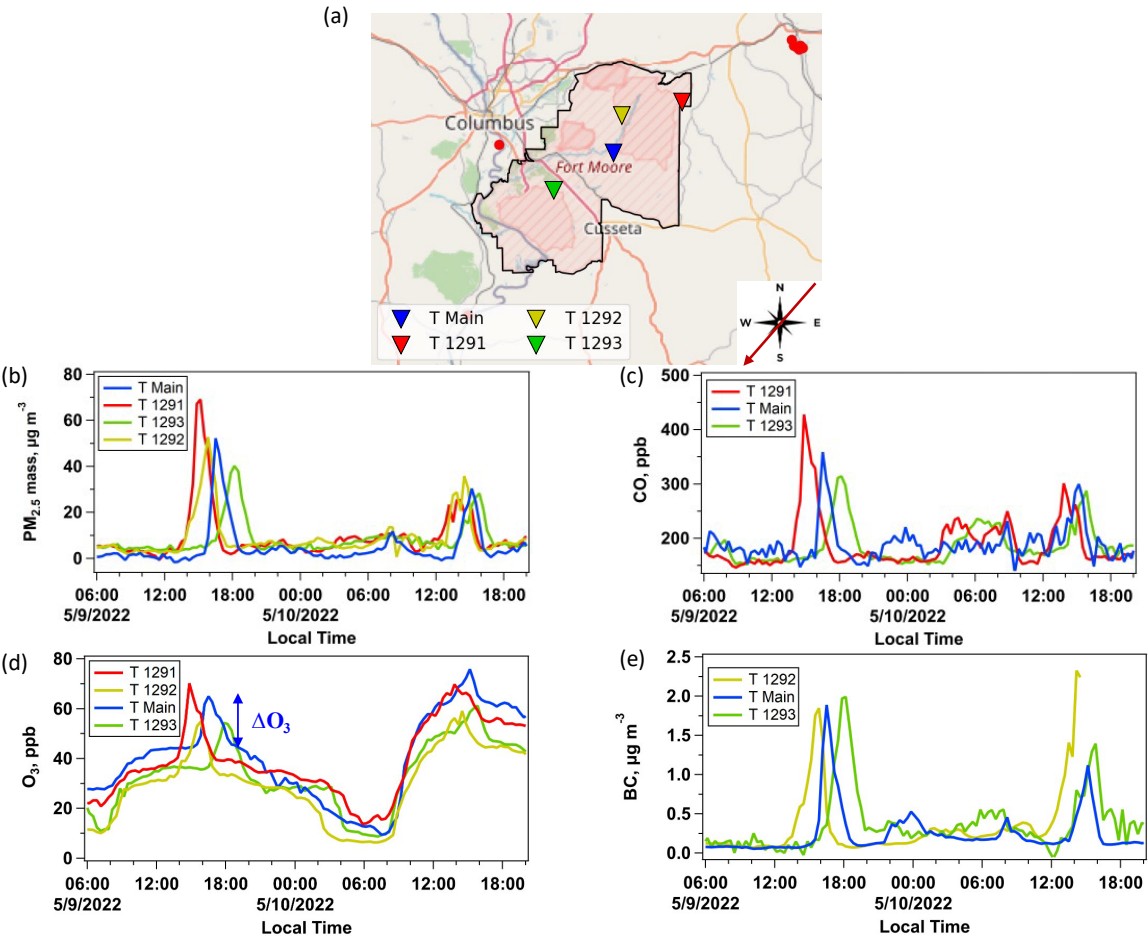


**Figure 8.** Case study of smoke detection sequentially at 4 monitoring trailers. (a) Map of the Fort showing historical satellite data
from the FIRMS website observed for May 9, 2022 (satellite overpass happened on May 9, 2022 at 12:38, 13:54, and 14:42) and
average wind vector from 13:00 to 16:00 local time. Time series showing 20 minutes data of (b) $PM_{2.5}$ mass and (c) CO on main
trailer, (d) $O_3$ concentration, and (e) BC concentration for, main trailer, T1291, T1292, and T1293. Note that no CO instrument was
operating at T1292 and no BC data for T1291.


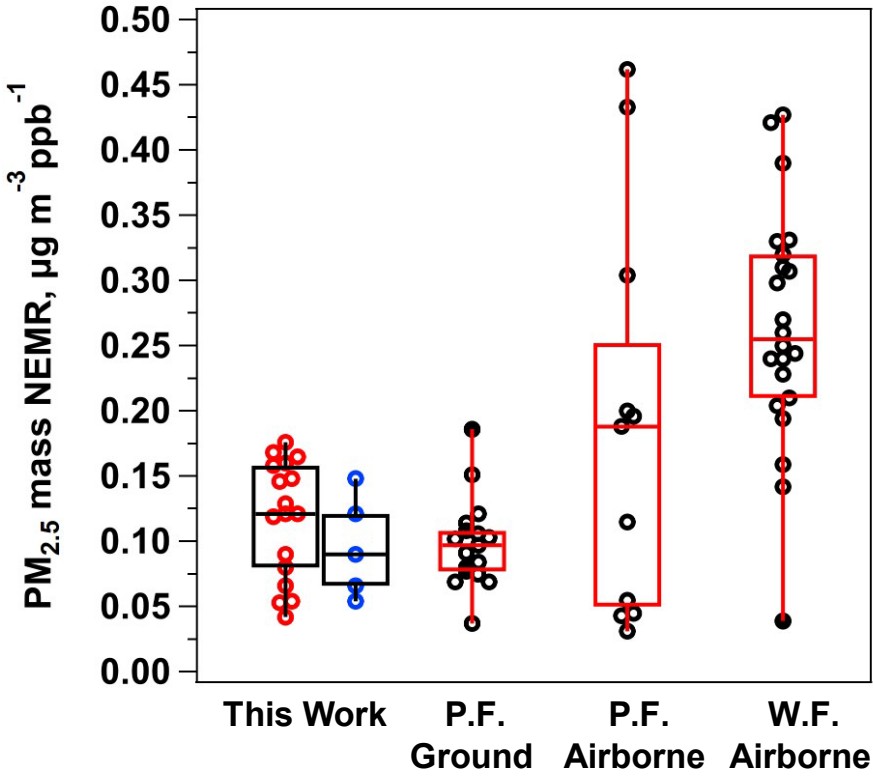


**Figure 9.** Box plot of PM$_{2.5}$ mass NEMRs of smoke events of estimated age ≤ 1 hour in this study in comparison to other studies.
Blue symbols are smoke plumes with observed O$_3$ enhancements. The horizontal line inside the box represents the median of the
data. The top line of the box represents the third quartile (Q3), and the bottom line represents the first quartile (Q1). Colored circles
represent data outliers. P. F. is Prescribed Fires, W.F. is wildfires. Some of the emission ratios reported in literature and included
in the plot correspond to ΔOA/ΔCO since OA tends to dominate ΔPM$_{2.5}$ mass concentration (see Table S10).


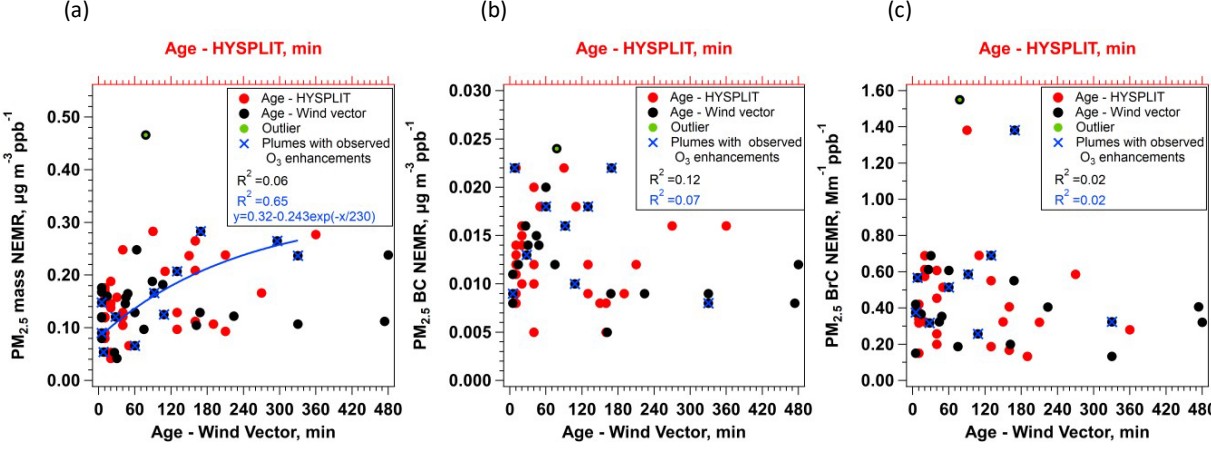


**Figure 10.** (a) PM$_{2.5}$ mass, (b) BC, and (c) BrC NEMRs of all studied smoke events as a function of age estimated using average
wind vector and HYSPLIT analysis. Smoke plumes with observed O$_3$ enhancements are identified. Linear regression coefficients
of determination (r$^2$) for all data and for just O$_3$ enhancement periods are identified. The exponential fit equation for PM$_{2.5}$ mass
NEMRs for O$_3$ enhancement periods is shown in (a).


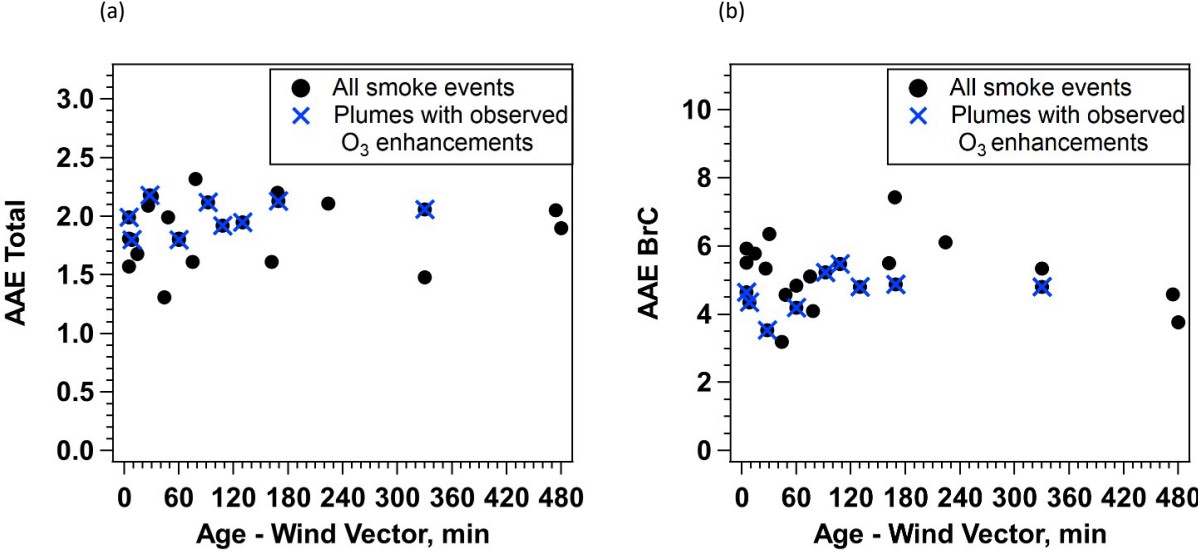


**Figure 11.** Average AAE values for a) total light absorption (BC+BrC) and b) BrC species for all smoke events for which
aethalometer data is available. Smoke plumes with observed $O_3$ enhancements are identified.




