# Peer review of "A Multi-site Passive Approach for Studying the Emissions and Evolution of Smoke from Prescribed Fires"

_EGUsphere, 2024_

## Referee Comment (RC2)

**Review of manuscript egusphere-2024-1485**

**A Multi-site Passive Approach 1 for Studying the Emissions and Evolution of Smoke from Prescribed Fires**

*The goal of this project is to study the emissions and evolution of smoke from prescribed fires and provide data to test model simulations.*

From conclusion: *This data set will be used to assess models predicting the impact of prescribed fires on air quality to enhance the use of prescribed burning in land management practices by minimizing impacts on populations*

**General Comments**

This paper describes an observational study of prescribed fire smoke conducted at Fort Moore in southeastern Georgia, US. Prescribed fire regularly is used on Fort Moore and in the region where the Fort is located. The stated goal of the study was to measure and characterize "the emissions and evolution of smoke from prescribed fires and provide data to test model simulations". The project deployed one ground-level, fixed location air monitoring station at Fort Moore during the 2021 prescribed fire season. During the 2022 prescribed fire season (Mar – May), four additional ground-level, fixed location sites were established. The monitoring sites measured CO, PM2.5, BC, BrC, O3 and NOx. (However, the NOx measurements are not reported in the paper.) The study utilizes meteorological observations from a Remote Automated Weather Station (RAWS) located on Fort Moore, Hysplit model trajectories driven by high-resolution (1-km inner domain) WRF simulations, Fort reports on fire activity, and satellite observations of active fires to determine the fires impacting the monitoring sites and estimate smoke age.

The paper reports emissions as normalized excess mixing ratios (NEMR), the excess mixing ratios of PM2.5, BC, and BrC normalized to CO, as is commonly done in biomass burning studies. The study focused on smoke plume events, defined as periods when 20-minute average PM2.5 levels > 35 ug/m3 and 40-minute PM2.5 levels > 30 ug/m3. A total of 64 smoke events were recorded across all sampling sites over the 2021 and 2022 deployments. This smoke event number is a count of sampling site level events, e.g. in 2022 a single fire could result in 5 smoke events if the smoke impacted all 5 sampling sites and resulted in PM2.5 levels meeting smoke event thresholds. For smoke events < 1 hour in age the NEMR are taken as emission ratios (ER) which represent the composition of fresh smoke. If concurrent CO2 measurements are available ER can be used to calculate emission factors (EF). The authors compare their ER results with previous, examine the variability of NEMR with smoke age, and identify and discuss instances of ozone enhancement. The authors also provide a summary of daytime versus nighttime ER, i.e. smoke measured during day when fires are presumed to be active versus smoke measured at night when fires are likely to be less active and probably smoldering dominated.

The observational dataset reported in the manuscript may be valuable for evaluating smoke dispersion and air quality model simulations of prescribed fires. However, the presentation of the data is incomplete, and it is unclear what crucial ancillary data may or may not be available for

model evaluation. I also have some concerns regarding the method used for assigning smoke age. employed. Additionally, I believe there is room for the authors to expand some on their analysis and discussion. Below, I elaborate on these comments and provide a few suggestions to the authors for strengthening the paper. Then I provide specific comments, followed by recommendations for technical corrections.

Study goal(s). What kind of models will be tested? The authors do not describe the types of models for which their study is intended to provide evaluation data nor how their study's observational dataset could be used to evaluate these models. The uses of the ambiguous target models are not discussed. In the introduction the authors need to address the following: what are the target model(s), what are these models used for, in what ways do these models need improvement and/or evaluation, and how will this study's observational dataset meet these model evaluation needs and potential lead to improved models. This information is crucial for determining how the study results meet the study purpose: "The goal of this project is to study the emissions and evolution of smoke from prescribed fires and provide data to test model simulations."

A critical component to making this data useful for model evaluation is knowing the fire ignition times and, when possible, the duration of active fire. Also, more specific info on the type vegetation and fuels that were burned and when the previousprescribed burns took place. This information can be useful in estimating fueling loading, which is needed for bottom-up estimates of emission fluxes which are need for model evaluation.

For fire events detected by MODIS/VIIRS, verifying that the preceding satellite overpasses had a clear view (not clouds) of the site but did not detect active fire, can help narrow the window of possible fire ignition time (e.g. Fig S7).

Is it possible to obtain more detailed information from Fort Moore on the prescribed fire events such as ignition time, ignition method, ignition duration, unit area? This information would be very useful, perhaps even necessary, to use the smoke event observations for model evaluation, which is the stated goal of this project.

Data collected. It is quite unfortunate the study did not include $CO_2$ measurements. High quality $CO_2$ mixing ratios can be easily measured using LiCOR $CO_2$ instruments. These instruments are inexpensive, easy to maintain and operate, and are designed for extended duration, unattended atmospheric measurements as were conducted in this study. $CO_2$ measurements are used to calculate modified combustion efficiency, MCE (Urbanski et al. 2022) an index of the relative mix of the flaming and smoldering emissions in a smoke sample. MCE is widely used to compare burning conditions between emissions studies, between fires, and temporally for individual fires. MCE can be very useful in understanding why ER vary among different studies or different fires within study. The lack of MCE measurements somewhat limits what can be learned from this study and how the finding might be applied in modeling studies. Further, $CO_2$ measurements would have allowed for calculations of emission factors, EF (Urbanski et al. 2022) (The long-term background $CO_2$ measurements would have allowed the authors to account for diel and seasonal variations in background $CO_2$.) Having EF would have enabled a

more comprehensive and enlightening comparison of the current study with previous studies. Further, EF are needed for bottom-up emission flux calculations that are input to smoke dispersion and air quality models. To use the study dataset in smoke dispersion or air quality models, the authors will need to estimate, based on other studies, what the EFCO are in order to derive EFPM2.5 and EFBC for calculating emission flux rates of these pollutants. Also, for simulating O3 or SOA formation processes, VOC emissions profiles are needed. MCE can be very is useful for estimating VOC emissions when EF for VOC were not measured. The authors will be unable to do this easily due to the absence of $CO_2$ measurements.

Even without $CO_2$ measurements, the study has solid potential to improve understanding and modeling of emissions and smoke transport for prescribed fires in the Southeastern US. However, the description and inventory of data available for this study is incomplete and the presentation needs to be improved. This needed to better demonstrate how the study addresses the study goals of elucidating emissions and evolution of smoke from prescribed fires and providing data for model evaluation.

Additional data. The following ancillary data would be of great value, and in some cases is necessary, to utilize the smoke measurements for a robust evaluation of smoke dispersion or air quality models.

Fort More fires, if available:

- Burn unit – Area of unit burned, centroid of burn unit or burned area polygon, forest/vegetation type and fuel type information, including if mechanical fuel treatments were conducted on site, and date of or time since previous prescribed fire.
- Burn conditions - ignition start time, ignition method, and end time of ignition, and fuel moistures

Satellite data – MODIS/VIIRS overpass times, number of active fire detections per pass and FRP for all fires that impacted the monitoring sites. Including overpass that did not record an active fire but occurred during the time-period in which a fire generated emissions that impacted monitoring sites is useful as well along with cloud cover info from imagery or RAWS and Colombus airport observations.

Smoke age – the reported smoke age needs to be clarified as discussed below in specific comments. However, once clarified, including the estimated emission time or time window for each smoke event would be of great use in addition to estimated smoke age alone.

I recommend the authors compile expanded tables in the supplement that includes as much of the above information as possible for each smoke event.

**Specific Comments**

**L28-42**: See the surveys of prescribed fire use prepared by the Coalition of Prescribed Fire Councils and the National Association of State for specific info on acres of prescribed fire use in the Southeastern US (https://www.stateforesters.org/newsroom-category/publications/) :

**L43-52**: The authors should provide background on the use and air quality impacts of prescribed fire in the Southeast US. See for example: Afrin & Garcia-Menendez, 2020; Larkin et al., 2020; Bian et al., 2020.

**L53**: "Both wildfires and prescribed fires emit a large variety of gases and particulates (Liu et al., 2017b; Burling et al., 2011)."

Update references with Gkatzelis et al. (2024); Permar et al. (2021) (wildfire) and Travis et al. (2023) (prescribed fire).

**L56**: "PM2.5, (particulate matter with aerodynamic diameter of 2.5 micrometers or smaller), is directly emitted as primary particles and also formed from condensation of emitted gases and their oxidation products, where a major component is secondary organic aerosol (SOA) (Liu et al., 2016; May et al., 2014)."

Needs revision. 1) SOA is not always a major component of aged biomass burning PM or necessarily the primary fate of emitted organic gases and 2) the volatile nature of organic PM must be mentioned, especially the fact that both primary and secondary PM can evaporate, reducing PM as a plume ages.

**L59-61**: "PM2.5 exposure has been linked in many epidemiological studies to serious health problems such as respiratory, cardiovascular, and neurological diseases, as well as increased risk of adverse birth outcomes (Liu et al., 2015; Reid et al., 2016; Naeher et al., 2007)."

Recommend one or two newer sources and perhaps a couple studies related specifically to wildland fire smoke and maybe even prescribed fire smoke.

**L63-71**. This section could use some cleaning-up. E.g., brief overview of active fire detection, burned area, and FRP to estimate fire location, burned area, fuel consumption, and emissions.

**L72-L74**: "Aircraft (fixed wing and helicopters) and more recently drones are commonly used in airborne studies of wildland fires (Decker et al., 2021b; Cubison et al., 2011) and have been deployed for prescribed burning studies (Yokelson et al., 1999; May et al., 2014; Pratt et al., 2011)."

Recommend UAS references, e.g. Aurell et al 2021; 2023

**L79-86**: See Fiddler et al 2024 and refs therein re: FIRE-EX AQ ground-based mobile emissions measurements.

**L87-91**: Authors should note there are several studies where wildfire smoke fortuitously impacted atmospheric chemistry labs or pre-existing monitoring locations, see as example Selimovic et al. 2019. Also consider noting that regulatory air monitoring sites, despite their limited measurement suites, can be very valuable in studies of smoke impacts (see any of several pubs by Dan Jaffe from U Wash.).

**L104-116**: This is a very good description and informative background on the Fort. Is it possible to include a description of tree species that dominate the uplands and bottomlands? Also, is prevention of wildfires the primary purpose of prescribed burning or are there also ecological objectives e.g., restoration/maintenance of longleaf pine, etc.?

**L138-139**: "Calibration of CO analyzers was performed before and after each field study using a 100 ppm CO in air standard purchased from nexAir (Memphis, TN)"

100 ppm seems quite high for ambient air monitoring. Did you verify the instrument precisions, accuracy, and stability (e.g. pre- and post-burn-period) with CO standards typical of ambient air and the CO levels observed in the smoke events during your study?

**L140-142**: "$O_3$ was measured using an ultraviolet (UV) photometric analyzer (Thermo Fisher Scientific Inc, model 49C, Franklin, MA)"

Is this an instrument found to have artifacts in smoke impacted environments? See: https://research.fs.usda.gov/treesearch/63344

**L151-152**: "The TEOM is a US-EPA approved instrument for measuring the mass concentration of ambient PM2.5 and PM10 (Liu et al., 2017a)."

Please note TEOM model number. Also, I believe the authors mean that the TEOM used may also be used for Federal Equivalent Method (FEM) regulatory measurements: https://www.epa.gov/sites/production/files/2019-08/documents/designated_reference_andequivalent_methods.pdf

**L153-154**: "The sample air is preconditioned to a temperature of 50 °C to remove liquid water interferences."

Please confirm that these are typical/recommended operating flow temperature. Interesting considering Pagonis et al. (2023) who found that when smoke samples were "heated to 40–45 °C in an airborne thermal denuder, 19% of lofted smoke PM1 evaporates"

**L161-162**: "Regional hourly PM2.5 mass was reported at two Environmental Protection Division (EPD) sites."

Please note the PM2.5 measurement techniques employed at these regulatory monitoring sites.

**L179**: Can the authors cite a good BrC review for the readers?

**L178-195**: Why not use 370 nm for BrC determination? What have previous studies done?

**L228-229**: "Also, small or relatively cool fires may not be detected, especially when there is significant cloud coverage or thick smoke"

The authors should note that for prescribed understory burns in forests, the focus of this study, a continuous, thick forest canopy may also obstruct satellite detection.

**L217-230**: It's unclear why they need active fire detections to identify location and timing of prescribed fires on the base. Did the authors not have access to the location, size, start and end times of prescribed fires conducted on the base during their study?

**L243-251**: Unfortunate all TEOM were not inter-compared as this can identify an outlier unit. Collocation at an ambient air monitoring site is a good way to gain an additional check instrument performance.

**L275**: Should provide definition of 'peaks' here.

**L277-281**: For 2022, please also note the number of days on which when PM smoke events were observed.

**L282**: "We focus on the larger smoke plumes…"

This should be rephrased. The focus is on smoke events with PM peaks of > 35 ug/m3 for 20-min mean and > 30 ug/m3 for a 40-minute mean. Such events could result from a plume "small" in volume (or a fire "small" in size) but concentrated due to location of fire, transport conditions, and emission production rates.

**L289 – 290**: For comparison, please note the correlation between PM2.5 and CO, BC, and BrC for no-event periods. Also, clarify if non-events data points include all observations during the entirety of the measurement periods.

**L295-296**: "When a smoke plume is identified the goal is to link it to a specific burn area and determine the transport time."

This is also important for using the dataset to evaluate smoke transport/dispersion models.

**L296-297**: "…we had limited beforehand information on the timing and location of planned burns."

Does the research team have access (after the fact) to the location (burn unit centroids), unit area, and ignition time of the burns? This is critical information that should be easily obtained from the Fort's land management / fire management team.

**L352-358. Smoke age determination. It is unclear how to interpret the smoke age.**

**This needs to be clarified.**

For the example given in Fig. 5, the duration of the three smoke events are 8 hours, 14 hours, and 8 hours. For all three events, there are changes in wind speed and direction of the 8 -14 hours of the events. Why were the trajectories shown in Fig 5. initiated at the times selected? The trajectory start times relative to the start of each smoke event are ~ 2:40 h 4/7, ~0 h on 4/7, and ~3 h on 4/8. Also, temporally, what wind observations were used in the Fig 5 analysis? Given the long duration of the events and accompanying variability of the winds, the smoke age will vary over the event. If the authors want to report the average smoke age for the events, the age should be based on some temporal averaging across the events' durations. The authors need to describe what exactly they believe their "smoke age" is intended to represent.

**L364**: "The substantial difference between modeled and observed winds suggests that relying on the wind vector based on observed winds is more reliable in this instance."

Please provide reasoning for this statement. The mere existence of a substantial difference between the two methods does not suggest the observed wind method is more reliable than the model wind method.

Did the authors consider using the Colombus airport receptor site as an additional comparison of the observed wind and hysplit model approaches to some age?

Also, comparison of the WRF model winds versus the Colombus airport wind observations may provide some insight the model performance.

**L368-373**: Forest canopy can have significant effects on winds (see Mallia et al. 2018; Huang et al. 224)

**L415-420**: In instances like this, when the source fire is very close to the receptor, transport near the surface may be heavily influenced by fire – atmosphere interactions. It is very possible that neither the RAWS (located some distance away) or the WRF simulations will be a good indicator of atmospheric flows this close to an active fire.

**L420-435**: The authors should note that the emission production rate and fire convective energy/plume rise would also be expected to have a large impact on the concentration measured

downwind. In the case of the April 4, 2022, nighttime event it is likely the smoke was produced by a mostly smoldering fire and released into the atmosphere with very little convective energy and was trapped in the shallow PBL.

**L430-432**: "The much higher PM2.5 mass concentrations measured on April 4, 2022 suggests that the trailer received a more direct smoke hit on that day than on February 11, 2022 or February 12, 2022, despite the fire being closer on February 11 and having a very similar distance to the one detected on February 12."

**L459-460**: "The ages of the smoke detected based on wind vector analysis were 266, 296, 330 and 480 minutes, for the various trailers"

Are the differences in smoke age between the trailers consistent with the observed difference in the smoke arrival time?

**L480-502**:

The ERPM2.5 reported in Yokelson et al. 2011; Yokelson et al. 2009; Akagi et al. 2012, and Burling et al. 2011 were based on nephelometry measurements. Later analysis of aerosol mass spectrometer data taken on some of the same flights (May et. al 2014), found that nephelometry-based measurements did not provide a reliable measure of PM2.5. I recommend removing these datasets from your discussion. Doing so may impact your assessment of differences between airborne & ground-based measurements on prescribed fires.

I also recommend including Travis et al. (2023) in this discussion as the study includes many Southeastern U.S. prescribed fires.

**L498-502**: The discussion of aerosol mass from evaporation of semi-volatile aerosol particle components is important. It should be noted that plume dilution can also lead to evaporation, perhaps offsetting the cooling effect of a lofted plume (e.g. May et al. 2013; May et al. 2015; Sinha et al. 2022). I encourage the authors to consult these studies as well as Pagonis et al (2023) and consider expanding this discussion.

For fresh smoke, BC to PM2.5 ER and NOx to PM2.5 ER would be interesting as they may provide information about fire behavior, e.g. yields of BC and NOx per unit mass of fuel consumed would be lower for smoldering dominated fire compared with flaming dominated fires.

**The April 21, 2022, event seems very interesting**. What about FRP? Fuel type? Burning conditions? Can the authors expand the discussion of this event?

**L547-548**: "…PM2.5 mass concentration NEMR consistently increases with physical age (r2=0.65), evidence of secondary aerosol formation through a photochemical process that directly involve O3…"

This doesn't provide evidence that the process resulting in increased PM2.5 NERM results from a photochemical process that **directly involves O3**. Please propose a mechanism and/or site appropriate studies to support this statement.

**L550 – 577**:

The ERPM2.5 reported in Yokelson et al. 2011; Yokelson et al. 2009; Akagi et al. 2012 were based on nephelometry measurements. Later analysis of aerosol mass spectrometer data taken on some of the same flights (May et. al 2014), found that nephelometry-based measurements did not provide a reliable measure of PM2.5. I recommend removing these datasets from your discussion. Please refer to May et al. 2014 re: the aerosol emissions originally reported in the Akagi et al. 2012 study.

Please include Garofalo et al. (2019) and Pagonis et al. (2023), these papers are very relevant to this discussion.

**L581-583**: "Satellite images do not show any visible differences in vegetation between the forested areas burnt on and off the base. Additionally, no further information regarding the fuel types in the off-base lands could be obtained."

Please note what satellite imagery was used. Also, the authors could consult recent vegetation maps (e.g. LANDFIE,  https://www.landfire.gov/ ) which may provide a more sophisticated assessment of the vegetation (e.g. forest type) and potential fuel loading (https://www.landfire.gov/fuel) than is possible from visual inspection of imagery by a non-expert.

**L604-612**: It *may* be worth including the lab study of Selimovic et al. (2018) in this discussion, something for the authors to consider.

**615-616**: "The method was successful in capturing a significant number of smoke events (64)."

Include the number of days on which smoke events were successfully measured.

**617**: change 'determined' to 'estimated' (the two methods used at times yielded very different smoke age estimates)

**L617-618**: Would be useful to include the number of fires for which smoke events were measured.

**Figures**

**Maps need scale bars!**

**Figure 1**. Nice figure, but please add scale bar.

**Figure 3**. Regression line in missing in panel b.

**Figure 10**. Since wind vector and hysplit are on same scale (as they should be) maybe use just one set of labels (smoke age) for the x-axis. Is the ozone enhancement plotted versus wind vector age or hysplit age? Please clarify in caption.

**Supplement**

**Fig. S6**. Was all or most of the area within the burn units (gray shaded polygons) burned? Please add a scale bar to 6a. What is the native and displayed resolution of the Colombus airport and PCSG school monitoring sites? In tables S3 & S7, please state the reason for the missing CO observations.

**Table S8**. Please clarify in table heading that listed overpass times are local time.

**Technical**:

L132: suggested text change to: "used as a tracer of smoke movement and dispersion". Also, it may be worth noting that CO is used as a standard tracer of combustion sources in atmospheric chemistry studies.

L334: "from the northwest, west, and southwest" change to "westerly" ?

L614-615: "We describe a ground-based measurement method for characterizing smoke from prescribed fires based on continuous monitoring at multiple sites for an extended period in a regularly burned region."

Suggest changing 'ground-based measurement method' to 'ground-based observational study'

L620:  maybe change "conservative parameter" to "conserved co-emitted species"?

L629: "indicating combined secondary O3 and particle mass formation" change to "indicating the formation of O3 and secondary organic aerosol"

**References**

Afrin, S., Garcia-Menendez, F. (2020). The influence of prescribed fire on fine particulate matter pollution in the Southeastern United States. Geophysical Research Letters, 47, e2020GL088988. https://doi.org/10.1029/2020GL088988

Aurell et al. (2023) Seasonal emission factors from rangeland prescribed burns in the Kansas Flint Hills grasslands, Atmospheric Environment, Volume 304, 2023, 119769,ISSN 1352-2310, https://doi.org/10.1016/j.atmosenv.2023.119769.

Aurell et al. (2021) Wildland fire emission sampling at Fishlake National Forest, Utah using an unmanned aircraft system, Atmospheric Environment, Volume 247, 2021, 118193, ISSN 1352-2310, https://doi.org/10.1016/j.atmosenv.2021.118193.

Bian, Q., Ford, B., Pierce, J. R., & Kreidenweis, S. M. (2020). A decadal climatology of chemical, physical, and optical properties of ambient smoke in the western and southeastern United States. Journal of Geophysical Research: Atmospheres, 125, e2019JD031372. https://doi.org/10.1029/2019JD031372

Fiddler, M. N., Thompson, C., Pokhrel, R. P., Majluf, F., Canagaratna, M., Fortner, E. C., et al. (2024). Emission factors from wildfires in the Western US: An investigation of burning state, ground versus air, and diurnal dependencies during the FIREX-AQ 2019 campaign. Journal of Geophysical Research: Atmospheres, 129, e2022JD038460. https://doi.org/10.1029/2022JD038460

Garofalo et al. (2019) Emission and Evolution of Submicron Organic Aerosol in Smoke from Wildfires in the Western United States. ACS Earth and Space Chemistry 2019 3 (7), 1237-1247 DOI: 10.1021/acsearthspacechem.9b00125

Hung, W.-T., Campbell, P. C., Moon, Z., Saylor, R., Kochendorfer, J., Lee, T. R., & Massman, W. (2024). Evaluation of an in-canopy wind and wind adjustment factor model for wildfire spread applications across scales. Journal of Advances in Modeling Earth Systems, 16, e2024MS004300. https://doi.org/10.1029/2024MS004300

Larkin, N. K., Raffuse, S. M., Huang, S., Pavlovic, N., Lahm, P., & Rao, V. (2020). The Comprehensive Fire Information Reconciled Emissions (CFIRE) inventory: Wildland fire emissions developed for the 2011 and 2014 U.S. National Emissions Inventory. Journal of the Air & Waste Management Association, 70(11), 1165–1185. https://doi.org/10.1080/10962247.2020.1802365

Mallia, Derek V.; Kochanski, Adam K.; Urbanski, Shawn P.; Lin, John C. 2018. Optimizing smoke and plume rise modeling approaches at local scales. Atmosphere. 9: 166. https://research.fs.usda.gov/treesearch/56156

May et al. 2013. Gas-particle partitioning of primary organic aerosol emissions: 3. Biomass burning, JGR 119, 11327-11338. https://doi.org/10.1002/jgrd.50828

Pagonis et al. Impact of Biomass Burning Organic Aerosol Volatility on Smoke Concentrations Downwind of Fires Environ. Sci. Technol. 2023, 57, 17011−17021 https://doi.org/10.1021/acs.est.3c05017

Permar, W., Wang, Q., Selimovic, V., Wielgasz, C., Yokelson, R. J., Hornbrook, R. S., et al. (2021). Emissions of trace organic gases from western U.S. wildfires based on WE-CAN aircraft measurements. Journal of Geophysical Research: Atmospheres, 126, e2020JD033838. https://doi.org/10.1029/2020JD033838

Selimovic, V., Yokelson, R. J., Warneke, C., Roberts, J. M., de Gouw, J., Reardon, J., and Griffith, D. W. T.: Aerosol optical properties and trace gas emissions by PAX and OP-FTIR for

laboratory-simulated western US wildfires during FIREX, Atmos. Chem. Phys., 18, 2929–2948, https://doi.org/10.5194/acp-18-2929-2018, 2018.

Selimovic, V., Yokelson, R. J., McMeeking, G. R., and Coefield, S.: In situ measurements of trace gases, PM, and aerosol optical properties during the 2017 NW US wildfire smoke event, Atmos. Chem. Phys., 19, 3905–3926, https://doi.org/10.5194/acp-19-3905-2019, 2019.

Sinha, A., I. George, A. Holder, W. Preston, M. Hays, AND A. Grieshop. Development of Volatility Distribution for Organic Matter in Biomass Burning Emissions. Environmental Science: Atmospheres. Royal Society of Chemistry, Cambridge, Uk, , 0000, (2022). https://doi.org/10.1039/D2EA00080F

Travis, K. R., Crawford, J. H., Soja, A. J., Gargulinski, E. M., Moore, R. H., Wiggins, E. B., et al. (2023). Emission factors for crop residue and prescribed fires in the Eastern US during FIREX-AQ. Journal of Geophysical Research: Atmospheres, 128, e2023JD039309. https://doi.org/10.1029/2023JD039309

Urbanski, Shawn P.; O'Neill, Susan M.; Holder, Amara L.; Green, Sarah A.; Graw, Rick L. 2022. Emissions. In: Peterson, David L.; McCaffrey, Sarah M.; Patel-Weynand, Toral, eds. 2022. Wildland Fire Smoke in the United States: A Scientific Assessment. Cham, Switzerland: Springer Nature Switzerland AG. 121-165. Chapter 5. https://doi.org/10.1007/978-3-030-87045-4_5.

---

## Author Comment (AC1)

**Reviewer 1**

We thank the reviewer for insightful comments. We have addressed these comments point by point below and revised the manuscript accordingly. The reviewer comments are in blue italics, our responses are in black text, and changes made to the manuscript are in red text. Line numbers refer to the original manuscript.

We first provide some context for this research and the structure of this paper. The measurements were made at a US military base that required permission for access, which in our case, was controlled by the base fire management team. During the performance of the study, fire managers raised questions about the data. This paper addresses many of those questions, resulting in a detailed analysis of why we trust our data, how we identified the fires responsible for the smoke measured and distinguished them from burning in the surrounding region, and why we did not detect smoke from some of the fires set on the base. We view these as valid questions that are pertinent to people who practice prescribed burning with the goal to minimize adverse impacts of their burning and which can be under significant scrutiny. Despite significant past collaborations with the fire team at this location, our access to the base was ultimately ended prior to our planned completion of the study.

**General comments**

*Regarding the results discussed in the light of the international literature - most of the ref are from the US and there are a few from Australia and Africa -all related to savanna burning – which is an odd choice of references considering the study sites were "pine-dominated uplands, and hardwood-dominated bottomlands' relate more to temperate forests than savannas. There are plenty of references for temperate southern Australian forests – would be more beneficial to compare studied emission factors/ ratio/ concentrations with relevant international literature.*

Response: We have reviewed the literature and added a reference that is complementary to our analysis. We looked at studies for temperate southern Australian forests; many lacked crucial information that would allow comparisons with our results, such as no PM data (Guérette et al., 2018), missing key information necessary for PM ER calculation (Possell et al., 2015), or a focus exclusively on specific types of vegetation (Reisen et al., 2018). However, we have included a recent study on prescribed fires in the eastern US, which examines forests with vegetation similar to those in our study (Travis et al., 2023). Overall, we have included a wide range of burning locations (fuels), such as savannas, to demonstrate that the variability we observe within our data is greater than the variability in the type of material burned based on literature values.

*I found ms a bit too long with too many minor details (at least for readers outside the USA or W Georgia specific area), I would prefer to read a more condensed version, less figures (there are 10 figures!) but a more prominent story. I kept going back and forward to make sure I keep track of the story.*

Response: While we acknowledge the reviewer's comment regarding the paper's length and details, we feel the comprehensive nature of the study can address concerns and issues that fire

authorities may have about monitoring emissions and attributing impacts of their burning. Example questions are: how do you know that smoke was from a fire we set, why do you not record smoke from some of the fires we set and why is the amount of smoke recorded variable? How do you assess background levels of measured species if not making measurements immediately upwind of the fire? What is the quality of your data and how do you assess this? This paper aims to answer these types of questions and provide a detailed analysis of burning from one region and will hopefully be useful to both scientific and fire management audiences. See our general first comment above. We have modified the last paragraph of the introduction to hopefully make this point. We have tried to clarify the flow of the paper by improving section labels and in some cases added introductory lines to sections.

*L360-361 and elsewhere reduce ' respectively' – 'wind data are 75, 14, and 162 minutes for April 6 (blue shading), 7 (yellow shading) and 8 (green shading), 2021 events respectively." - and just state number and date eg '75 min for April 6 (blue shading), 14 min for 7 April (yellow shading)" etc – it is a bit frustrating to stop and re-read which respectively corresponds to which number*

Response: The indicated lines and other lines were rephrased as follows:

L360-361: For the three events discussed in Fig. 5, physical ages estimated using the wind vector averaged from observed RAWS wind data are 75 minutes, 14, and 162 minutes for April 6 (blue shading), 14 minutes for April 7 (yellow shading), and 162 minutes for April 8 (green shading), 2021 events respectively.

L361-362: For the same events and using HYSPLIT trajectories closest to the surface and passing through the identified sources, ages were estimated as 130 minutes for April 6 (blue shading), 10 minutes for April 7 (yellow shading), and 40 minutes for April 8 (green shading) respectively.

*Technical corrections*

*Please use SI units for area – ha not acres (at least provide ha in the brackets)*

Response: Units in hectares (ha) have been added in brackets throughout the manuscript.

*L108 – 'unintended wildfires' -can a wildfire be intended?*

Response: This is a good point. The word "unintended" was omitted from the sentence.

*L131-132- not sure what it means – 'used as a smoke tracer to track its movement and dispersion' -to track smoke movements or CO movement? Since its= CO. Did you mean 'used as a tracer to track smoke movement ..'?*

*Also 1 month is not really a long-lived species*

Response: The lines indicated were rephrased to better address the idea, incorporating the second reviewer's comments on those same lines.

L131-132: Carbon monoxide serves as a standard tracer for combustion sources in atmospheric chemistry studies since it is a relatively long-lived species, with a typical lifetime of ~ 1 month, emitted during incomplete combustion and used as a  tracer of  smoke  movement and dispersion (Forrister et al., 2015; Liu et al., 2016).

*L362 – I am lost which 'respectively' you refer to "130, 10, and 40 minutes respectively" - 75,14,162? or blue or yellow or green or all together?*

Response: As per the previous comment, this line was also changed to be as follows:

L361-362: For the same events and using HYSPLIT trajectories closest to the surface and passing through the identified sources, ages were estimated as 130 minutes for April 6 (blue shading), 10 minutes for April 7 (yellow shading), and 40 minutes for April 8 (green shading) .

*L488 – 'proscribed'- prescribed?*

Response: The typo was corrected in the manuscript.

**Other changes made in the manuscript, not directly related to any reviewer comments, are minor adjustments aimed at enhancing the clarity and structure of our sentences.**

---

## Author Comment (AC2)

**Reviewer 2**

We thank the reviewer for insightful comments. We have addressed these comments point by point below and revised the manuscript accordingly. The reviewer comments are in blue italics, our responses are in black text, and changes made to the manuscript are in red text. Line numbers refer to the original manuscript.

We first provide some context for this research and the structure of this paper. The measurements were made at a US military base that required permission for access, which in our case, was controlled by the base fire management team. During the performance of the study, fire managers raised questions about the data. This paper addresses many of those questions, resulting in a detailed analysis of why we trust our data, how we identified the fires responsible for the smoke measured and distinguished them from burning in the surrounding region, and why we did not detect smoke from some of the fires set on the base. We view these as valid questions that are pertinent to people who practice prescribed burning with the goal of minimizing adverse impacts of their burning and which can be under significant scrutiny. Despite significant past collaborations with the fire team at this location, our access to the base was ultimately ended prior to our planned completion of the study.

*Comments:*

*Study goal(s). What kind of models will be tested? The authors do not describe the types of models for which their study is intended to provide evaluation data nor how their study's observational dataset could be used to evaluate these models. The uses of the ambiguous target models are not discussed. In the introduction the authors need to address the following: what are the target model(s), what are these models used for, in what ways do these models need improvement and/or evaluation, and how will this study's observational dataset meet these model evaluation needs and potentially lead to improved models. This information is crucial for determining how the study results meet the study purpose: "The goal of this project is to study the emissions and evolution of smoke from prescribed fires and provide data to test model simulations."*

**Response:**

Thanks for pointing out the ambiguous reference to modeling in the manuscript. The collected data can be used to evaluate ground-level pollutant concentrations predicted by "smoke" models in prescribed fire simulations. There are a variety of smoke models ranging in scale from local to regional. Local scale models typically focus on fire propagation, emissions, smoke plume formation and its dispersion in the near field. Regional scale models on the other hand start with less detailed emissions and plume-rise parameterizations but include chemical transformations and long-range transport over much larger domains. Our concentration data cover measurements over a large range of distances from the burn plots. The fresh plume measurements with ages less than 1 hour can be used in evaluating the predictions of local scale models. Examples of such models are Wildland urban interface Fire Dynamics Simulator (WFDS) (Mell et al., 2007) and

QUIC-Fire (Linn et al., 2020). They can also be used in evaluating the emissions and plume-rise parameterizations of larger scale models. The BlueSky framework (Larkin et al., 2009) includes several such parameterizations. More aged samples can be used in evaluating the predictions of downwind concentrations in coupled fire-atmosphere models such as WRF-SFIRE (Mandel et al., 2011) as well as chemical transport models such as Community Multiscale Air Quality (CMAQ) model (Appel et al., 2021), when they are equipped with fire plume parameterizations.

We have added this information in the manuscript as follows:

L92-101:  Here, we present results  from a two-year study utilizing fixed monitoring stations and continuous sampling in a region of active prescribed burning at Fort Moore in central Georgia, USA .  The observations are analyzed to identify smoke plumes and determine their sources, such as those set within the Fort or from burning in surrounding areas. We also use these data to estimate the age of the smoke detected to determine emission ratios and changes with plume age  of $PM_{2.5}$ mass, BC, and BrC and their variability. Not all smoke from the prescribed fires set within the Fort are  detected so the overall impact of all fires on regional air quality cannot be determined and is better addressed by a model simulation. Instead, our goal is to sample multiple smoke events so that an analysis of the data will provide a robust characterization of smoke from prescribed burning within the Fort and in the region and sufficient data  to evaluate ground-level pollutant concentrations predicted by "smoke" models in prescribed fire simulations. Our concentration data cover measurements over a large range of distances from the burn plots. Fresh plume measurements with ages less than 1 hour can be used in evaluating the predictions of local scale models such as the Wildland urban interface Fire Dynamics Simulator (WFDS) (Mell et al., 2007) and the QUIC-Fire (Linn et al., 2020). They can also be used in evaluating the emissions and plume-rise parameterizations of larger scale models like the BlueSky framework (Larkin et al., 2009). Additionally, more aged smoke measurements can be used to test the predictions of downwind concentrations in coupled fire-atmosphere models such as WRF-SFIRE (Mandel et al., 2011) as well as chemical transport models like the Community Multiscale Air Quality (CMAQ) model (Appel et al., 2021), when they are equipped with fire plume parameterizations.  to evaluate ground-level pollutant concentrations predicted by "smoke" models in prescribed fire simulations. Our concentration data cover measurements over a large range of distances from the burn plots. Fresh plume measurements with ages less than 1 hour can be used in evaluating the predictions of local scale models such as the Wildland urban interface Fire Dynamics Simulator (WFDS) (Mell et al., 2007) and the QUIC-Fire (Linn et al., 2020). They can also be used in evaluating the emissions and plume-rise parameterizations of larger scale models like the BlueSky framework (Larkin et al., 2009). Additionally, more aged smoke measurements can be used to test the predictions of downwind concentrations in coupled fire-atmosphere models such as WRF-SFIRE (Mandel et al., 2011) as well as chemical transport models like the Community Multiscale Air Quality (CMAQ) model (Appel et al., 2021), when they are equipped with fire plume parameterizations. In the following sections, we describe the methodology, data analysis approach, case studies of various detected or missed smoke plumes so that attribution of

smoke from fires within the Fort can be assessed., and initial f Findings on emission estimates of PM2.5 mass, BC, and BrC and their evolution are compared to other prescribed and wildfire studies. These findings are can help needed to assess the impact of prescribed burns by a specific entity or organization on a variety of public health and policy issues.

*A critical component to making this data useful for model evaluation is knowing the fire ignition times and, when possible, the duration of active fire. Also, more specific info on the type vegetation and fuels that were burned and when the previous prescribed burns took place. This information can be useful in estimating fueling loading, which is needed for bottom-up estimates of emission fluxes which are need for model evaluation.*

**Response:**

We agree that this information would be useful and have added as much information as we had from the fire managers, but we were limited by the extent of information they provided, and the level of the information decreased as the study progressed ending in our access being restricted to the base. Thus, in many instances, we had to rely only on publicly available data. See the response at the end of this comment.

*For fire events detected by MODIS/VIIRS, verifying that the preceding satellite overpasses had a clear view (not clouds) of the site but did not detect active fire, can help narrow the window of possible fire ignition time (e.g. Fig S7).*

We agree that this approach effectively narrows the window of possible fire ignition times; however, some uncertainty will remain. Therefore, we relied on running back trajectories instead of forward ones. In our work, all fire times based on MODIS/VIIRS data were confirmed by the absence of fires on the preceding satellite overpass. Nevertheless, analyzing available satellite data, particularly cloud coverage during preceding passes, is valuable for modelers when ignition time is critical for running their models.

*Is it possible to obtain more detailed information from Fort Moore on the prescribed fire events such as ignition time, ignition method, ignition duration, unit area? This information would be very useful, perhaps even necessary, to use the smoke event observations for model evaluation, which is the stated goal of this project.*

**Response:**

See the comment above about limited data from Fort Moore.

*Data collected. It is quite unfortunate the study did not include CO2 measurements. High quality CO2 mixing ratios can be easily measured using LiCOR CO2 instruments. These instruments are inexpensive, easy to maintain and operate, and are designed for extended duration, unattended atmospheric measurements as were conducted in this study. CO2 measurements are used to calculate modified combustion efficiency, MCE (Urbanski et al. 2022) an index of the relative mix of the flaming and smoldering emissions in a smoke sample. MCE is widely used to compare burning conditions between emissions studies, between fires, and temporally for individual fires. MCE can be very useful in understanding why ER vary among different studies or different fires*

*within study. The lack of MCE measurements somewhat limits what can be learned from this study and how the finding might be applied in modeling studies. Further, $CO_2$ measurements would have allowed for calculations of emission factors, EF (Urbanski et al. 2022) (The long-term background $CO_2$ measurements would have allowed the authors to account for diel and seasonal variations in background $CO_2$.) Having EF would have enabled a more comprehensive and enlightening comparison of the current study with previous studies. Further, EF are needed for bottom-up emission flux calculations that are input to smoke dispersion and air quality models. To use the study dataset in smoke dispersion or air quality models, the authors will need to estimate, based on other studies, what the $EF_{CO}$ are in order to derive $EF_{PM2.5}$ and $EF_{BC}$ for calculating emission flux rates of these pollutants. Also, for simulating $O_3$ or SOA formation processes, VOC emissions profiles are needed. MCE can be very is useful for estimating VOC emissions when EF for VOC were not measured. The authors will be unable to do this easily due to the absence of $CO_2$ measurements.*

*Even without $CO_2$ measurements, the study has solid potential to improve understanding and modeling of emissions and smoke transport for prescribed fires in the Southeastern US. However, the description and inventory of data available for this study is incomplete and the presentation needs to be improved. This needed to better demonstrate how the study addresses the study goals of elucidating emissions and evolution of smoke from prescribed fires and providing data for model evaluation.*

*Additional data. The following ancillary data would be of great value, and in some cases is necessary, to utilize the smoke measurements for a robust evaluation of smoke dispersion or air quality models.*

*Fort More fires, if available:*

- *Burn unit – Area of unit burned, centroid of burn unit or burned area polygon, forest/vegetation type and fuel type information, including if mechanical fuel treatments were conducted on site, and date of or time since previous prescribed fire.*

- *Burn conditions - ignition start time, ignition method, and end time of ignition, and fuel moistures*

*Satellite data – MODIS/VIIRS overpass times, number of active fire detections per pass and FRP for all fires that impacted the monitoring sites. Including overpass that did not record an active fire but occurred during the time-period in which a fire generated emissions that impacted monitoring sites is useful as well along with cloud cover info from imagery or RAWS and Colombus airport observations.*

*Smoke age – the reported smoke age needs to be clarified as discussed below in specific comments. However, once clarified, including the estimated emission time or time window for each smoke event would be of great use in addition to estimated smoke age alone.*

*I recommend the authors compile expanded tables in the supplement that includes as much of the above information as possible for each smoke event.*

**Response:** We have compiled an Excel file with all the data that we were able to obtain from the fire management team at Fort Moore to address the issues raised above. The file is included in the supplementary material. For each of the 64 smoke events studied in the paper, whenever data is available, the file includes:

- The detection site and its latitude and longitude
- The local start and end time of smoke detection at the monitoring site
- The fire/smoke source
- The total and burn area (in acres)
- The latitude and longitude of the centroid of the burn area
- The date of the last burn of the burning area
- The stand condition, which is the only data we have about vegetation
- Whether or not the wildfires in the same year reduced burn area acres
- Ignition time
- Ignition type

We were not able to obtain the duration of active fire.

Fuel moisture data are available online through RAWS USA Climate Archive at https://raws.dri.edu/index.html.  The closest RAWS weather station to all sites is named Ft. Benning Georgia.

Cloud coverage data are available online through https://worldview.earthdata.nasa.gov.

Satellite data – including MODIS/VIIRS overpass times, the number of active fire detections per pass, and FRP for all fires that impacted the monitoring sites, can be downloaded from the FIRMS website at https://firms.modaps.eosdis.nasa.gov/. This includes overpasses that did not record an active fire but occurred during the time period when fire-generated emissions impacted the monitoring sites. Users can specify spatial limits based on the longitudes and latitudes provided in the attached Excel file.

The reader is now directed to all the abovementioned data by the following changes in the manuscript:

L196-197: In our analysis, we used meteorological and fuel moisture data from the Remote Automated Weather Stations (RAWS) available online (https://raws.dri.edu/index.html).  The closest RAWS weather station to all sites is named Ft. Benning Georgia (Fig. 1a).

L228-230: Also, small or relatively cool fires may not be detected, especially when there is significant cloud coverage or thick smoke. Cloud coverage data are available online (https://worldview.earthdata.nasa.gov) and satellite data, including MODIS/VIIRS overpass times, the number of active fire detections per pass, and FRP for all fires that impacted the monitoring sites, can be downloaded from the abovementioned FIRMS website. Burn data provided by Fort Moore were used with the FIRMS data to minimize limitations with each method for identifying sources of observed smoke. For each of the 64 smoke events studied in the paper, burn data are added to the supplementary material (Table S1).

We agree with the reviewer for emphasizing the importance of $CO_2$ measurements. We were planning to add $CO_2$ in the final deployment, but that was canceled due to denial of access to Fort Moore.

*L28-42: See the surveys of prescribed fire use prepared by the Coalition of Prescribed Fire Councils and the National Association of State for specific info on acres of prescribed fire use in the Southeastern US (https://www.stateforesters.org/newsroom-category/publications/) :*

*2021 Survey*

*2020 Survey*

*2018 Survey*

**Response:** We have added pertinent information in the suggested references above as follows:

L28-42: Large and intense wildfires have been increasing over the past few decades  and their emissions are a critical concern (Singleton et al., 2019; Jaffe et al., 2020). Fire is also an essential ecological process and prescribed burning, which is the act of starting controlled fires for specific purposes, is an important tool for restoration of ecosystems, land management, and reducing fuel to prevent destructive wildfires (Kelp et al., 2023). Prescribed fires are typically conducted during favorable conditions associated with the fuel type and amount, soil moisture, and meteorology.  For example, in 2018, the United States Department of Agriculture (USDA) Forest Service indicated a high risk of hazardous wildfires over approximately 234 million acres (~ 95 million ha) of forest lands in the US (Wyden and Manchin, 2020). However, prescribed fires were conducted over approximately 8.5 million forestry/rangeland acres (~ 3.4 million ha)  in 2018 (Melvin, 2020) . The southeastern US has a long history of using prescribed fires (Melvin, 2021).For example, in 2017, 7.6 million acres (3 million ha) out of the 11.3 million acres (4.6 million ha) burned nationally were in the southeast (Melvin 2018). Florida and Georgia each exceeded 1 million acres (0.4 million ha) burned annually (Melvin, 2018).  ~~roughly 2.18, 1.26, and 0.94 million acres were burned in 2017 in Florida, Georgia, and Alabama respectively, representing almost two thirds of the prescribed burning conducted nationally (as published by Statista Research Department based on National Interagency Coordination Center in March 20, 2018). The National Interagency Coordination Center estimated that the total prescribed fire acres burned in 2017 nationally was 6.43 million acres.~~ Recognizing the need to mitigate the size and severity of wildfires, prescribed burning is anticipated to increase in the coming years (USDA, 2022).

*L43-52: The authors should provide background on the use and air quality impacts of prescribed fire in the Southeast US. See for example:  Afrin & Garcia-Menendez, 2020; Larkin et al., 2020; Bian et al., 2020.*

**Response:** The indicated lines have been changed to:

L43-52: While prescribed burning can be performed under favorable weather conditions, it can still contribute to serious local and regional air pollution as it is a source of primary and secondary air pollutants (Lee et al., 2008). Like other types of biomass burning, prescribed burning releases  large amounts of particulate matter, CO, and inorganic and organic compounds (Lee et al., 2005), which have negative effects on health and visibility (Bell, 2004; Huang et al., 2019). Particularly in the southeastern US, prescribed burning was significantly associated with high $PM_{2.5}$ levels (Afrin and Garcia-Menendez, 2020; Larkin et al., 2020). Prescribed fires are often conducted at urban-rural interfaces creating a buffer zone to prevent the spread of wildfires towards the built environment. However, this means that the planned fires often occur closer to populated areas, and potentially lead to high population exposure due to this proximity. Although prescribed fires generally produce less pollutants by consuming less fuel per area burned than wildfires, the population health costs can be substantially higher for prescribed fires due to burning near higher population densities (Borchers-Arriagada et al., 2021).

*L53: "Both wildfires and prescribed fires emit a large variety of gases and particulates (Liu et al., 2017b; Burling et al., 2011)."*

*Update references with Gkatzelis et al. (2024); Permar et al. (2021) (wildfire) and Travis et al. (2023) (prescribed fire).*

**Response:** References are updated in the manuscript as suggested.

*L56: "PM2.5, (particulate matter with aerodynamic diameter of 2.5 micrometers or smaller), is directly emitted as primary particles and also formed from condensation of emitted gases and their oxidation products, where a major component is secondary organic aerosol (SOA) (Liu et al., 2016; May et al., 2014)."*

*Needs revision. 1) SOA is not always a major component of aged biomass burning PM or necessarily the primary fate of emitted organic gases and 2) the volatile nature of organic PM must be mentioned, especially the fact that both primary and secondary PM can evaporate, reducing PM as a plume ages.*

**Response:** The revised version of L56 becomes: $PM_{2.5}$, (particulate matter with aerodynamic diameter of 2.5 micrometers or smaller), is directly emitted as primary particles and also formed from condensation of emitted gases and their oxidation products,  (Liu et al., 2016; May et al., 2014). While secondary organic aerosol (SOA) can be a significant component of aged biomass burning $PM_{2.5}$, its contribution changes depending on emissions and atmospheric conditions. Additionally, the volatile nature of primary and secondary organic components of $PM_{2.5}$ can lead to evaporation and a net loss in mass as the plume ages.

*L59-61: "PM2.5 exposure has been linked in many epidemiological studies to serious health problems such as respiratory, cardiovascular, and neurological diseases, as well as increased risk of adverse birth outcomes (Liu et al., 2015; Reid et al., 2016; Naeher et al., 2007)."*

*Recommend one or two newer sources and perhaps a couple studies related specifically to wildland fire smoke and maybe even prescribed fire smoke.*

**Response:** References are updated in the manuscript as suggested.

*L63-71. This section could use some cleaning-up. E.g., brief overview of active fire detection, burned area, and FRP to estimate fire location, burned area, fuel consumption, and emissions.*

**Response:** The indicated lines has been changed to:

L63-71: Detection and characterization of wildland fires is an important step towards assessing their impacts. Remote sensing via satellites can detect wildland fires by thermal anomalies (Kuenzer et al., 2008) or vegetation changes (Mildrexler et al., 2007). While satellite-based approaches offer valuable insights (Martinsson et al., 2022; Ichoku and Kaufman, 2005; Christopher et al., 1998), challenges such as cloud cover, spatial resolution limitations, and the complex nature of fire emissions can hinder accurate detection and quantification of fire impacts, especially for lower-intensity fires like prescribed burns (Liu et al., 2019; Wang et al., 2018; Martin et al., 2018). Therefore, factors like Fire Radiative Power (FRP), burned area estimation, and fuel consumption modeling are often integrated into fire monitoring systems (Li et al., 2020; Nguyen and Wooster, 2020). ~~Satellite-based approaches are highly useful (Ichoku and Kaufman, 2005; Christopher et al., 1998; Kaufman et al., 1989; Martinsson et al., 2022), but there are limitations, including temporal coverage and spatial resolution (both horizontal and vertical) (Liu et al., 2019; Ichoku et al., 2016), detector sensitivity, interferences (e.g., clouds or surface conditions), and interactions between different emissions (Liu et al., 2019; Wang et al., 2018; Martin et al., 2018). This can lead to significant under-detection of fires and limitations in quantifying emissions needed to determine population exposures, especially for lower intensity prescribed fires (Nowell et al., 2018; Larkin et al., 2020; Martin et al., 2018; Buysse et al., 2019; Jaffe et al., 2020).~~

*L72-L74: "Aircraft (fixed wing and helicopters) and more recently drones are commonly used in airborne studies of wildland fires (Decker et al., 2021b; Cubison et al., 2011) and have been deployed for prescribed burning studies (Yokelson et al., 1999; May et al., 2014; Pratt et al., 2011)."*

*Recommend UAS references, e.g. Aurell et al 2021; 2023*

**Response:** References are updated in the manuscript as suggested and the indicated lines has been changed to:

L72-L78: Aircraft (fixed wing and helicopters) and more recently drones are commonly used in airborne studies of wildland fires (Decker et al., 2021b; Cubison et al., 2011; Aurell and Gullett, 2024) and have been deployed for prescribed burning studies (Yokelson et al., 1999; May et al., 2014; Pratt et al., 2011; Aurell et al., 2021). Airborne studies provide high spatial resolution data that are often used to assess evolution of smoke properties by measurements at various downwind distances, however, it is non-continuous, and can miss certain aspects of smoke emissions, such as longer-term smoldering, especially at night (Burling et al., 2011). Employing a combination of airborne and ground-based measurements can be beneficial in providing a

comprehensive view of the plume (Burling et al., 2011; Akagi et al., 2014; Yokelson et al., 2013; Strand et al., 2016).

*L79-86: See Fiddler et al 2024 and refs therein re: FIRE-EX AQ ground-based mobile emissions measurements.*

**Response:** References are updated with the recommended sources and the following is added to:

L79-86: In ground-based studies,  mobile labs may capture dynamic air quality patterns and to some extent assess spatial variability of species  in plumes and their changes with plume age (Levy et al., 2014; Fiddler et al., 2024; Lee et al., 2023). However, they are usually limited in space and instrumentation capacity, such as filter samples collected only during stationary measurements (Warneke et al., 2023). Interferences from the power source, vibration and speed changes during transportation can affect instrument stability and performance leading to inaccurate measurements or limiting the type of instruments that can be used. Attempting to track wildland smoke plumes can be challenging due to unpredictable winds and dispersion conditions combined with access limitations. For example, Burling et al. reports successfully sampling smoke from 2 out of 14 prescribed fires using a battery powered mobile FTIR system (Burling et al., 2011).

*L87-91: Authors should note there are several studies where wildfire smoke fortuitously impacted atmospheric chemistry labs or pre-existing monitoring locations, see as example Selimovic et al. 2019. Also consider noting that regulatory air monitoring sites, despite their limited measurement suites, can be very valuable in studies of smoke impacts (see any of several pubs by Dan Jaffe from U Wash.).*

**Response:** The indicated lines have been changed to:

L87-91: Fixed ground-based monitoring stations equipped with various instruments provide continuous, localized measurements for short or long-term monitoring for studies assessing diurnal, seasonal, and long-term trends in air pollution. Multiple sites provide spatial coverage within a region. A variety of highly sensitive instruments can be deployed, ensuring accurate and precise measurements of various pollutants that can be compared with air quality data across different locations for regional assessments (Strand et al., 2016; Warneke et al., 2023). The importance of pre-existing fixed monitoring sites lies in their ability to capture wildfire smoke events that can occur at any time (Selimovic et al., 2019; Jaffe et al., 2022). These sites often include regulatory monitoring stations, which are highly valuable for studying local and regional smoke impacts over both short and long-term periods. For example, Jaffe et al. used $PM_{2.5}$ and CO observations from a regulatory monitoring site in Sparks, NV, collected from May to September between 2018 and 2021, as indicators of wildfire smoke in urban areas (Jaffe et al., 2022). Investigating emissions and evolution of prescribed fires based on fixed sites is not as common, and there are limitations with this approach, but also some advantages.

*L104-116: This is a very good description and informative background on the Fort. Is it possible to include a description of tree species that dominate the uplands and bottomlands? Also, is*

*prevention of wildfires the primary purpose of prescribed burning or are there also ecological objectives e.g., restoration/maintenance of longleaf pine, etc.?*

**Response:** The indicated lines have been edited as the following:

L104-116: Prescribed burning at Fort Moore Army Base, (formerly Fort Benning), in west central Georgia, United States, was studied during March through May of 2021 and February through May of 2022. Since 1981, prescribed burning has been used as a land management tool at the 182,000 acres military base, of which 145,000 acres are forested lands. Vegetation is characterized by pine-dominated uplands, and hardwood-dominated bottomlands, with the dominant tree species being longleaf pine and white oak, respectively. Unintended small wildfires ignited during military training exercises also occur at the base and the land managers have been recording data on both prescribed fires and wildfires since the 1980s. Prescribed burning at the Fort has been effective; it has reduced the frequency of wildfires from ~ 300-500 wildfires/year in the early 1980s to less than 100 wildfires/year in the mid-1990s. During this period the prescribed fire burnt area changed from ~7,500 acres in 1981 to ~ 12,000 acres in 1992. Currently, 30,000 woodland acres are burned annually using controlled fires, with a future planned burning of 45,000 acres annually. Prescribed burning on the Fort is also used for ecological objectives, such as restoring the longleaf pine forest and creating and maintaining habitat for red-cockaded woodpeckers. Prescribed burning occurs from December through May when there is sufficient but not excessive rainfall, and suitable temperatures and wind conditions to burn deadwood, brush, and low-growing vegetation accumulating on the forest floor. The area of the base is divided into 332 burn units that range in size from 100 to 1,800 acres and are burnt alternately every two to three years.

*L138-139: "Calibration of CO analyzers was performed before and after each field study using a 100 ppm CO in air standard purchased from nexAir (Memphis, TN)"*

*100 ppm seems quite high for ambient air monitoring. Did you verify the instrument precisions, accuracy, and stability (e.g. pre- and post-burn-period) with CO standards typical of ambient air and the CO levels observed in the smoke events during your study?*

**Response:** The calibration cylinder was 100 ppm, but a standard addition was performed at 2.2 ppm for calibration by diluting the calibration CO gas with the ambient air flow. The line has been clarified: Calibration of CO analyzers was performed at 2.2 ppm concentration before and after each field study using a 100 ppm CO in air standard purchased from nexAir (Memphis, TN).

*L140-142: "O3 was measured using an ultraviolet (UV) photometric analyzer (Thermo Fisher Scientific Inc, model 49C, Franklin, MA)"*

*Is this an instrument found to have artifacts in smoke impacted environments? See: https://research.fs.usda.gov/treesearch/63344*

**Response:**

We would like to mention that our $O_3$ NEMRs, although not reported in this paper, tend to be in the range reported by other investigators who sampled over a range of $O_3$ and VOC concentrations. Our range of $O_3$ NEMRs for smoke events studied in this paper is 0.021-0.148 ppb ppb$^{-1}$.

Additionally, after further literature review, we found that Bernays et al. in 2022, published a comment (https://doi.org/10.5194/amt-15-3189-2022) on the reference you provided pointing out some omissions. Mainly, the type of scrubber used was misidentified as manganese dioxide ($MnO_2$) when in fact it was manganese chloride ($MnCl_2$). This led to the inaccurate conclusion that all UV-based $O_3$ instruments with solid-phase catalytic scrubbers show positive artifacts, whereas previous research found this not to be the case when employing $MnO_2$ scrubber types, like Gao and Jaffe, 2017 (https://doi.org/10.1016/j.atmosenv.2017.07.007) .

We have added the following to the manuscript:

L140-143: $O_3$ was measured using an ultraviolet (UV) photometric analyzer (Thermo Fisher Scientific Inc, model 49C, Franklin, MA) zeroed through an $O_3$ scrubber in the instrument, with LOD of 1.0 ppb and averaging time of 20 seconds. The analyzer was calibrated before and after each field deployment using an $O_3$ calibrator (Thermo Fisher Scientific Inc, model 49C, Franklin, MA). We note that $O_3$ measured in smoke by this instrument may be overestimated due to interferences from VOCs (Long et al., 2021), but the instrument used has been found to be in agreement with a federal reference method (Gao and Jaffe, 2017).

*L151-152: "The TEOM is a US-EPA approved instrument for measuring the mass concentration of ambient PM2.5 and PM10 (Liu et al., 2017a)."*

*Please note TEOM model number. Also, I believe the authors mean that the TEOM used may also be used for Federal Equivalent Method (FEM) regulatory measurements: https://www.epa.gov/sites/production/files/2019-08/documents/designated_reference_andequivalent_methods.pdf*

**Response:** We have added the model number to L151 in the manuscript, and we have clearly stated the ability of the TEOM to be used for FEM regulatory measurements based on its listing in the Designated Reference and Equivalent Methods.

L151:152: The TEOM series 1400a developed originally by Rupprecht & Patashnick is a US-EPA approved instrument for measuring the mass concentration of ambient $PM_{2.5}$ and $PM_{10}$ and could be used for Federal Equivalent Method (FEM) regulatory measurements (Liu et al., 2017a; Patashnick and Rupprecht, 1991).

*L153-154: "The sample air is preconditioned to a temperature of 50 °C to remove liquid water interferences."*

*Please confirm that these are typical/recommended operating flow temperature. Interesting considering Pagonis et al. (2023) who found that when smoke samples were "heated to 40–45 °C in an airborne thermal denuder, 19% of lofted smoke PM1 evaporates"*

**Response:** While heating the sample to 50°C can cause evaporation of highly volatile compounds, it is necessary in order to minimize interferences from the evaporation and condensation of water onto the filter and to provide a stable and reproducible measurement (Patashnick and Rupprecht, 1991).

We have added the following to the manuscript:

L153-154: The sample air is preconditioned to a temperature of 50 °C to remove liquid water interferences (Patashnick and Rupprecht, 1991), which may lead to the evaporation of highly volatile $PM_{2.5}$ components, potentially underestimating the total mass concentration.

*L161-162: "Regional hourly PM2.5 mass was reported at two Environmental Protection Division*

*(EPD) sites."*

*Please note the PM2.5 measurement techniques employed at these regulatory monitoring sites.*

**Response:** We have added the $PM_{2.5}$ measurement techniques employed at these regulatory monitoring sites as follows:

L161-162: Regional hourly $PM_{2.5}$ mass was reported at two Environmental Protection Division (EPD) sites. In the following analysis we compare the $PM_{2.5}$ measured within the Fort to the EPD measurements at the Columbus Airport and Phenix City South Girard (PCSG) school shown on the map in Fig. 1a. At Columbus Airport, the Teledyne T640, which is based on broadline spectroscopy, is used, while the Met One BAM-1022 mass monitor is used in Phenix City, utilizing a beta attenuation technique.

*L179: Can the authors cite a good BrC review for the readers?*

**Response:** We have added citations to three comprehensive papers on BrC for the readers' reference.

L178-179: BrC is largely produced from biomass burning (Hecobian et al., 2010; Laskin et al., 2015; Yan et al., 2018; Fleming et al., 2020) and in the following analysis used as a unique indicator of biomass burning smoke.

*L178-195: Why not use 370 nm for BrC determination? What have previous studies done?*

**Response:** While BrC compounds absorb light over a broad range of wavelengths in the visible spectrum, their maximum absorption typically occurs between 300-400 nm. For simplicity, BrC levels are often characterized by light absorption at a single wavelength, especially in studies using solvent-extracted BrC methods, with 365 nm being the most common wavelength used (Zeng et al., 2022; Liu et al., 2013; Washenfelder et al., 2015). We plan to compare this BrC

measurement to soluble BrC measured on filters collected during the study and so, for consistency, we use 365 nm.

*L228-229: "Also, small or relatively cool fires may not be detected, especially when there is significant cloud coverage or thick smoke"*

*The authors should note that for prescribed understory burns in forests, the focus of this study, a continuous, thick forest canopy may also obstruct satellite detection.*

**Response:** The line is changed to: Also, small or relatively cool fires may not be detected, especially when there is significant cloud coverage,  thick smoke, or a continuous, thick forest canopy, which can block satellite detection of prescribed understory burns in forests.

*L217-230: It's unclear why they need active fire detections to identify location and timing of prescribed fires on the base. Did the authors not have access to the location, size, start and end times of prescribed fires conducted on the base during their study?*

**Response:** Unfortunately, due to poor communication from the fire management team, we had limited access to necessary information such as burn schedules, ignition times, and fire durations. The available data, including burning locations and details on burn units, was received after the studies had already begun, impacting our analysis. This made satellite data essential for identifying smoke from other sources, such as prescribed burns and wildfires in the surrounding region.

*L243-251: Unfortunate all TEOM were not inter-compared as this can identify an outlier unit. Collocation at an ambient air monitoring site is a good way to gain an additional check instrument performance.*

**Response:** Inter-comparisons of all TEOM units for quality assurance would have been ideal. This required locating all instruments at a common site and recording smoke events. As noted, COVID restrictions at the start of the study and later limited access to the base made this difficult to achieve.

*L275: Should provide definition of 'peaks' here.*

**Response:** A definition to peak is added to the indicated line as follows:

L275-276: During this period, peaks of measured species were observed, as shown in the time series of $PM_{2.5}$ mass in Fig. 2b. A peak of a measured species is defined as the highest value observed within the data points, spanning from an initial rise until a return to background levels.

*L277-281: For 2022, please also note the number of days on which when PM smoke events were observed.*

**Response:** The information was added to manuscript as follows:

L278-281: In 2022, over the course of the entire burning season, 32 days recorded a total of 53 $PM_{2.5}$ mass concentration peaks greater than 35 $\mu g\ m^{-3}$  across the five measuring

sites, as shown in Fig. 2c with similar high concentrations, reaching 841 µg m$^{-3}$ for 20-minute averaged and 513 µg m$^{-3}$ for hourly-averaged data (Tables S54 to S87).

*L282: "We focus on the larger smoke plumes…"*

*This should be rephrased. The focus is on smoke events with PM peaks of > 35 ug/m3 for 20-min mean and > 30 ug/m3 for a 40-minute mean. Such events could result from a plume "small" in volume (or a fire "small" in size) but concentrated due to location of fire, transport conditions, and emission production rates.*

**Response:** The sentence is rephrased:

L282-287: We focus on  smoke plumes with higher PM$_{2.5}$ mass concentrations to identify their sources and estimate the emissions and evolution of PM$_{2.5}$ mass because the burning areas are readily identified (e.g., detected remotely by satellite) and the plume can be easily delineated from the background.

*L289 – 290: For comparison, please note the correlation between PM2.5 and CO, BC, and BrC for no-event periods. Also, clarify if non-events data points include all observations during the entirety of the measurement periods.*

**Response:** We added to the noted lines the correlation between PM$_{2.5}$ and CO, BC, and BrC and a clarification to what non-events data points are. The lines became:

L289-290: The linear relation between PM$_{2.5}$ and CO, BC, and BrC during events resulted in an $r^2$ of 0.85, 0.68, and 0.71 respectively. On the other hand, for non-events data, which include all observations during the entirety of the measurement period, $r^2$ drops to 0.12, 0.33, and 0.17 for PM$_{2.5}$ mass vs CO, BC, and BrC respectively.

*L295-296: "When a smoke plume is identified the goal is to link it to a specific burn area and determine the transport time."*

*This is also important for using the dataset to evaluate smoke transport/dispersion models.*

**Response:** For a better statement on the goals of the following sections the lines indicated were rephrased to:

 To study the emission and evolution of smoke plumes and make our measurements useful for evaluating smoke transport and dispersion models, we aim to link identified smoke plumes to specific burn areas and determine their transport time. Attribution of the smoke to specific fires is also useful for assessing the impacts of a specific prescribed burning program, such as the one at Fort Moore. Identifying the location of prescribed fires was complicated by several factors. In this study, we had limited beforehand information on the timing and location of planned burns from the burn managers.

*L296-297: "…we had limited beforehand information on the timing and location of planned burns."*

*Does the research team have access (after the fact) to the location (burn unit centroids), unit area, and ignition time of the burns? This is critical information that should be easily obtained from the Fort's land management / fire management team.*

**Response:** After the study, the fire management shared data including dates of burns, burning locations and details on burn units that we added to the supplementary material. Unfortunately, we were unable to obtain fire ignition times, except for specific burns that we are modeling, and the duration of active fire, even after requesting this information.

*L352-358. Smoke age determination. It is unclear how to interpret the smoke age.*

*This needs to be clarified.*

*For the example given in Fig. 5, the duration of the three smoke events are 8 hours, 14 hours, and 8 hours. For all three events, there are changes in wind speed and direction of the 8 -14 hours of the events. Why were the trajectories shown in Fig 5. initiated at the times selected? The trajectory start times relative to the start of each smoke event are ~ 2:40 h 4/7, ~0 h on 4/7, and ~3 h on 4/8. Also, temporally, what wind observations were used in the Fig 5 analysis? Given the long duration of the events and accompanying variability of the winds, the smoke age will vary over the event. If the authors want to report the average smoke age for the events, the age should be based on some temporal averaging across the events' durations. The authors need to describe what exactly they believe their "smoke age" is intended to represent.*

**Response:**

For the HYSPLIT simulations, we initially ran the backward trajectory from the start time of the smoke event. However, due to uncertainties in the WRF simulated winds, particularly at night when wind speeds are low, the backward trajectory occasionally missed the source. To address this, we conducted a series of HYSPLIT simulations with 20-minute intervals from the event start time until the source of smoke could be identified (when HYSPLIT trajectories intersect with a known fire). The 20-minute interval was chosen based on the temporal resolution of the WRF data. For instance, we first simulated the HYSPLIT at the event start time (01:00 local time) on April 6th, 2021 (Figure 1, b) but the trajectory did not intersect with any fires. We continued running HYSPLIT simulations with different start times until the trajectory intersected with the burn unit (at 04:40) as shown in the manuscript in Figure 5. The trajectory that

intersected the burn unit at ground level was then used to determine the smoke age.

[Figure]

*Figure 1.HYSPLIT back trajectories with different simulation start times during the smoke event on April 6, 2021. The trajectories in Figure (a) start at 1:00 local time, which is the start time of the event. The trajectories in Figure (b) start at 3:20 local ti time, which is the time when PM$_{2.5}$ reaches the highest concentration during the smoke event period.*

For smoke calculated using the wind vector, the average wind vector during the hour prior to detecting the peak (defined as the highest value observed within the data points spanning from an initial rise until a return to background levels) is first combined with the distance between the fire and the monitors to calculate the smoke age. If smoke age exceeds 1 hour, iteration by averaging the wind vector for the two hours leading to the peak is done. The latter step is repeated until convergence of the age calculated with the duration at which smoke was averaged. Thus in this method we get the average over the expected smoke transport time. This is explained by an example age calculation written in the supplementary material. This introduces uncertainty regarding the age of the smoke monitored before and after the peak (maximum concentration), particularly when the smoke event duration (from the start to the end of smoke monitoring) is prolonged, and when wind conditions are highly variable. While it is not possible to calculate this uncertainty, we note this in the manuscript as follows:

L352-358: An estimate of the smoke age is needed to separate fresh from aged smoke to estimate emissions of various species (i.e., in fresh smoke) and the changes in their concentrations with plume age. The physical age of smoke is the time it takes the smoke to be transported from the source to the monitoring sites. Following the concept presented for source identification, the transport time of smoke is estimated by averaging wind speed over the period it takes for the smoke to travel from the fire to the measurement sites, determined by iteration (mean wind speed recalculated with new transport time, until convergence). When the average wind speed in the hour leading up to the peak does not result in a smoke age of one hour or less, we begin iterative steps by calculating the average wind vector for additional one hour increments at a time. A detailed example on using average wind vector in estimating the physical age of smoke is provided in the Supplemental section S.1. It is important to note the uncertainty in the estimated smoke age using this method for smoke monitored before and after the peak (maximum concentration), particularly when the smoke event duration (from the start to the end of smoke monitoring) is prolonged, and when wind conditions are highly variable. The age was also determined from the HYSPLIT back trajectories as the time when the lowest trajectory intersects the source of smoke identified. The backward trajectory is initiated from the start time of the

smoke event. Due to uncertainties in the WRF simulated winds, particularly at night when wind speeds are low, the backward trajectory occasionally missed the source. Therefore, a series of HYSPLIT simulations with 20-minute intervals from the event start time until the source of smoke could be identified were conducted. The 20-minute interval was chosen based on the temporal resolution of the WRF data.

*L364: "The substantial difference between modeled and observed winds suggests that relying on the wind vector based on observed winds is more reliable in this instance."*

*Please provide reasoning for this statement. The mere existence of a substantial difference between the two methods does not suggest the observed wind method is more reliable than the model wind method.*

*Did the authors consider using the Colombus airport receptor site as an additional comparison of the observed wind and hysplit model approaches to some age?*

*Also, comparison of the WRF model winds versus the Colombus airport wind observations may provide some insight the model performance.*

**Response:** We thank the reviewer for their suggestion. A comparison with the observed winds at Columbus airport for the three events presented in Figure 5 was conducted and added as Figure S6 in the supplementary material. While the wind direction observed at the airport more closely aligns with that at the RAWS site in Fort Moore (though with faster winds at the airport, likely due to the forest canopy effect on wind flow), we agree that there is no conclusive evidence that the observed wind method is more reliable than the modeled wind method, as discussed later in the same section. Accordingly, the manuscript was edited as follows:

L364: The difference between modeled and observed wind for these three instances was further investigated by comparison with the observed wind at Columbus Airport.  As shown in Fig. S6, the wind direction observed at the airport aligns more closely with that observed at the RAWS site in Fort Moore (though with faster winds at the airport, likely due to the forest canopy effect on wind flow) than with the WRF modeled winds at both sites. However, it is difficult to determine which method is more reliable for studying any specific smoke event.

*L368-373: Forest canopy can have significant effects on winds (see Mallia et al. 2018; Huang et al. 224)*

**Response:** This is a good point, and it was highlighted in the manuscript as follows:

L370-372: For the wind vector analysis, observed winds are measured at one location and 2 meters above ground level with a single monitor, which  may not accurately represent the wind patterns along the entire smoke transport path, especially in forested areas where the canopy can affect the wind flow (Mallia et al., 2020).

*L415-420: In instances like this, when the source fire is very close to the receptor, transport near the surface may be heavily influenced by fire – atmosphere interactions. It is very possible that*

*neither the RAWS (located some distance away) or the WRF simulations will be a good indicator of atmospheric flows this close to an active fire.*

**Response:** This is a good point, and it was highlighted in the manuscript as follows:

L415-419: When several burning units are in close proximity and near the measurement site, identifying the specific source and smoke age can also be difficult (for example see Fig. S10). In this case burning in three units indicated by the Fort's Fire Management occurred at the same time close to each other and the trailer (distances of 0.6, 1.4, and 2.2 miles from the trailer). HYSPLIT trajectory at lowest altitude passes near (to the east), but not over the prescribed fires. Wind direction at the time of the event suggests influence  of a minor portion of the northern part of the fire. In such cases, transport near the surface may be heavily influenced by fire-atmosphere interactions, making it difficult to rely on data from RAWS or WRF simulations as accurate indicators of atmospheric flows close to an active fire.

*L420-435: The authors should note that the emission production rate and fire convective energy/plume rise would also be expected to have a large impact on the concentration measured downwind. In the case of the April 4, 2022, nighttime event it is likely the smoke was produced by a mostly smoldering fire and released into the atmosphere with very little convective energy and was trapped in the shallow PBL.*

**Response:** This is a good point, and it was highlighted in the manuscript as follows:

L431-435: The much higher $PM_{2.5}$ mass concentrations measured on April 4, 2022 suggests that the trailer received a more direct smoke hit on that day than on February 11, 2022 or February 12, 2022, despite the fire being closer on February 11 and having a very similar distance to the one detected on February 12. This can also be attributed to the much lower nighttime PBL on April 4, which was 9.8 m and caused all HYSPLIT trajectories to overlap as shown in Fig. 12d. Emissions from a smoldering fire with very little buoyant energy were most likely trapped in this shallow layer leading to high concentration measurements. During the daytime on February 11 and 12, higher PBL of 1645 and 1305 m, respectively, favored more vertical dispersion of smoke.

*L430-432: "The much higher PM2.5 mass concentrations measured on April 4, 2022 suggests that the trailer received a more direct smoke hit on that day than on February 11, 2022 or February 12, 2022, despite the fire being closer on February 11 and having a very similar distance to the one detected on February 12."*

**Response:** Yes, depending on the intensity of the burn, the rate of emissions, atmospheric stability, height of the smoke plume, the amount of dispersion and wind speed, measured concentrations may not be proportional to the distance from the burn.

*L459-460: "The ages of the smoke detected based on wind vector analysis were 266, 296, 330 and 480 minutes, for the various trailers"*

*Are the differences in smoke age between the trailers consistent with the observed difference in the smoke arrival time?*

**Response:** The difference in smoke age is close to the difference in peak arrival times. The peak (maximum PM$_{2.5}$ mass concentration) was observed at 15:02, 15:53, 16:25, and 18:16 on T1291, T1292, T1293, and T Main, respectively.

This information was added in the manuscript as follows:

L459-460: The ages of the smoke detected based on wind vector analysis were 266, 296, 330 and 480 minutes, for the various trailers. The difference in smoke age is close to the difference in peak arrival times with maximum PM$_{2.5}$ mass concentration observed at 15:02, 15:53, 16:25, and 18:16 on T1291, T1292, T Main, and T1293, respectively.

*L480-502:*

*The ER PM2.5 reported in Yokelson et al. 2011; Yokelson et al. 2009; Akagi et al. 2012, and Burling et al. 2011 were based on nephelometry measurements. Later analysis of aerosol mass spectrometer data taken on some of the same flights (May et. al 2014), found that nephelometry-based measurements did not provide a reliable measure of PM2.5. I recommend removing these datasets from your discussion. Doing so may impact your assessment of differences between airborne & ground-based measurements on prescribed fires.*

*I also recommend including Travis et al. (2023) in this discussion as the study includes many Southeastern U.S. prescribed fires.*

**Response:** Thank you for pointing this out. The indicated datasets were removed from the discussion. Figure 9 and Table S1 were changed accordingly. We also separated the first box in Figure 9 into two for improved clarity. The text in the manuscript is edited as follows:

L480-502: By focusing on fresh smoke (age less than 1 hour), the emissions ratios (ER) of the prescribed fires can be estimated and compared to those from other studies. The PM$_{2.5}$ mass concentration ER ranged between 0.04 and 0.18 µg m$^{-3}$ ppb$^{-1}$ and is shown in Fig. 9. These ERs are comparable to other prescribed fires measured at both ground level (Alves et al., 2010; Desservettaz et al., 2017; Korontzi et al., 2003; Balachandran et al., 2013) and aloft in airborne studies (Sinha et al., 2003; ; May et al., 2014; ; Gkatzelis et al., 2024; Travis et al., 2023) that span a large range of burning conditions and fuels (details are provided in Table S10). The mean PM$_{2.5}$ mass concentration ER for our data is $0.117 \pm 0.045$ µg m$^{-3}$ ppb$^{-1}$ and that of these other prescribed fire studies are $0.098 \pm 0.034$ µg m$^{-3}$ ppb$^{-1}$ for ground-based and $0.18$ $\pm 0.154$ µg m$^{-3}$ ppb$^{-1}$ for airborne measurements. There is substantial and similar variability in the ground-based measurements of prescribed fire ERs in this study relative to other studies.  More recent airborne-measured prescribed fires have reported substantially higher ERs (Fig. 9). Smoke transported for 10 minutes from the Blackwater river state forest prescribed fire reported by Gkatzelis et al. had an ER of 0.462 µg m$^{-3}$ ppb$^{-1}$ (Gkatzelis et al., 2024) and  Travis et al. reported a range of 0.188-0.433 µg m$^{-3}$ ppb$^{-1}$ for 22 prescribed fires studied

and grouped into 4 categories based on fuel type (Travis et al., 2023). Figure 9 also shows comparisons with wildfires reported in other studies (Liu et al., 2017b; Collier et al., 2016; Palm et al., 2020; Gkatzelis et al., 2024). Wildfire $PM_{2.5}$ mass ERs are significantly higher than ERs for prescribed fires in this work, with ER ranges between 0.04 and 0.43 $\mu$g m$^{-3}$ ppb$^{-1}$ and a mean of 0.264 ± 0.091 $\mu$g m$^{-3}$ ppb$^{-1}$ for wildfires, and the difference is statistically significant (two-tailed p value is < 0.0001). Lower $PM_{2.5}$ mass ERs from smaller prescribed fires has been noted in other studies (Liu et al., 2017b) and supports utilizing prescribed burning as a land management tool to limit wildfires. However, differences in altitude at which the measurements were made may have some effect on ERs. Selimovic et al. (Selimovic et al., 2019) noted that the $PM_{2.5}$/CO in ground-level smoke was about half of that observed from aloft apparently due to reduction in aerosol mass from evaporation of semi-volatile aerosol particle components resulting from higher surface temperatures compared to aloft. Pagonis et al. also found airborne OA NEMRs to be a factor of 2 higher than ground-based NEMRs giving the same interpretation (Pagonis et al., 2023). However, wWhen comparing ERs of prescribed fires in ground versus airborne studies of prescribed fires, shown in Fig. 9, the mean of airborne studies is a factor of ~ 1.9 higher than ground-based studies and the difference is statistically significant (p value is 0.025. difference is not significant (p value is 0.435).

*L498-502: The discussion of aerosol mass from evaporation of semi-volatile aerosol particle components is important. It should be noted that plume dilution can also lead to evaporation, perhaps offsetting the cooling effect of a lofted plume (e.g. May et al. 2013; May et al. 2015; Sinha et al. 2022). I encourage the authors to consult these studies as well as Pagonis et al (2023) and consider expanding this discussion.*

*For fresh smoke, BC to PM2.5 ER and NOx to PM2.5 ER would be interesting as they may provide information about fire behavior, e.g. yields of BC and NOx per unit mass of fuel consumed would be lower for smoldering dominated fire compared with flaming dominated fires.*

**Response:** Based on this comment and the previous one, the text in the manuscript was edited as follows:

L498-502: However, differences in altitude at which the measurements were made may have some effect on ERs. Selimovic et al. (Selimovic et al., 2019) noted that the $PM_{2.5}$/CO in ground-level smoke was about half of that observed from aloft apparently due to reduction in aerosol mass from evaporation of semi-volatile aerosol particle components resulting from higher surface temperatures compared to aloft. Pagonis et al. also found airborne OA NEMRs to be a factor of 2 higher than ground-based NEMRs giving the same interpretation (Pagonis et al., 2023). However, wWhen comparing ERs of prescribed fires in ground versus airborne studies of prescribed fires, shown in Fig. 9, the mean of airborne studies is a factor of ~ 1.9 higher than ground-based studies and the difference is statistically significant (p value is 0.025. difference is not significant (p value is 0.435).

The BC to $PM_{2.5}$ mass ratio in fresh plumes varies between 0.068 and 0.390, showing a clear negative correlation with $PM_{2.5}$ ER (r² = 0.69). This suggests that higher $PM_{2.5}$ ER is associated

with increased smoldering, resulting in a lower BC to PM$_{2.5}$ mass ratio. Due to the high discontinuity in the NO$_x$ dataset, it could not be included in the analysis.

*The April 21, 2022, event seems very interesting. What about FRP? Fuel type? Burning conditions? Can the authors expand the discussion of this event?*

**Response:** First, we identified a typo in the sentence describing this event, where the CO concentration was mistakenly reported in ppm instead of ppb. Moreover, to elaborate further on this event, the smoke from this event was traced to the only prescribed fire occurring at the Fort, located 5.6 miles from the measurement site at an azimuth of 155◦. Smoke peak was observed at ~ 13:40 when hourly wind speed reported was 3 mph at 117◦. The wind during the hour before was 6 mph at 148◦. Averaging the two hours would result in average wind vector of speed 4.3 mph and direction 138◦. The HYSPLIT trajectory does not show any intersection with the fire (Figure 2), even when run at different start times. Although the FRP reported on FIRMS does not differ from that of other fires, and there is no significant difference in vegetation type or fuel moisture, the most likely explanation for this event is that the smoke passing through the measurement site was not a direct hit but from the diluted boundary of the plume, which may have undergone photochemical processing, leading to high PM$_{2.5}$, BrC, and O$_3$ NEMRs.

The discussion is added to the manuscript as follows:

L537-540: This event corresponds to smoke from an identified prescribed fire at the Fort and has a relatively low ΔCO of 66.1  ppb, which is unexpected given the burn's proximity and the wind  speed on that day, causing ERs to be significantly higher. The HYSPLIT back trajectory from the measuring site does not intersect with the fire but passes close to it. Although the FRP reported on FIRMS does not differ from that of other fires, and there is no significant difference in vegetation type or fuel moisture, the most likely explanation for this event is that the smoke passing through the measurement site was not a direct hit but from the diluted boundary of the plume, which may have undergone photochemical processing, leading to higher PM$_{2.5}$, BrC, and O$_3$ NEMRs.)

[Figure]

*Figure 2. HYSPLIT back trajectories of smoke event on April 21, 2022 monitored at T1293 (green star).*

*L547-548: "...PM2.5 mass concentration NEMR consistently increases with physical age (r2=0.65), evidence of secondary aerosol formation through a photochemical process that directly involve O3..."*

*This doesn't provide evidence that the process resulting in increased PM2.5 NERM results from a photochemical process that **directly involves O3**. Please propose a mechanism and/or site appropriate studies to support this statement.*

**Response:** Thank you for pointing this out. Since we have no evidence to support this statement, the lines were revised into:

L546-548: Considering only smoke plumes in which $O_3$ enhancements were observed (i.e., smoke measured between 12:00 and 18:00), $PM_{2.5}$ mass concentration NEMR consistently increases with physical age ($r^2=0.65$), possibly evidence of secondary aerosol formation driven by photochemistry through a photochemical process that directly involve $O_3$, or some other related oxidant, (e.g., OH) (Liu et al., 2016).

*L550 – 577:*

*The ERPM2.5 reported in Yokelson et al. 2011; Yokelson et al. 2009; Akagi et al. 2012 were based on nephelometry measurements. Later analysis of aerosol mass spectrometer data taken on some of the same flights (May et. al 2014), found that nephelometry-based measurements did not provide a reliable measure of PM2.5. I recommend removing these datasets from your discussion. Please refer to May et al. 2014 re: the aerosol emissions originally reported in the Akagi et al. 2012 study.*

*Please include Garofalo et al. (2019) and Pagonis et al. (2023), these papers are very relevant to this discussion.*

**Response:** The specified references have been removed from the discussion, and the recommended ones have been added. The revised discussion lines are now as follows:

L550-577: A range of results for changes in $PM_{2.5}$ mass concentration NEMRs in wildland fires have been observed in other studies, including systematic increases, little change, or decreases with smoke age. To the best of our knowledge, no ground-based studies have been conducted on the evolution of smoke from prescribed fires, but frequent airborne studies have investigated prescribed and wildland smoke aging because of the ability to spatially characterize a single plume. While studying two prescribed fires in SC, May et al. (May et al., 2015) observed no statistically significant net change in OA NEMRs near the source and downwind for smoke transported for ≤ 1.5 hours. One of the two fires was studied for longer, and results showed downwind OA NEMRs over 2 to 5 hours of transport significantly lower than the NEMRs at the source, suggesting a net loss of emitted OA. Also, a decrease in OA/CO from 0.057 to 0.046 μg m$^{-3}$ ppb$^{-1}$ was reported after 4 hours in an airborne study of an 81 hectare prescribed fire in chaparral fuels on the central coast of California (Akagi et al., 2012). In contrast, an airborne study of 20 deforestation and crop residue fires on the Yucatan peninsula, reported an increase in

 For wildfires, Collier et al. (Collier et al., 2016) found increases, little change, and decreases with smoke age in different wildfire plumes measured in Oregon. For the selected large wildfires in the western US in summer, Palm et al. (Palm et al., 2020) reported that the OA NEMR remained almost constant at a value of ~ 0.25 µg m$^{-3}$ ppb$^{-1}$ as the plume aged from 20-50 minutes to 6 hours. In their analysis of the data of wildland fires studied during FIREX-AQ campaign in 2019, Pagonis et al. report OA NEMR increased from 0.2 g g$^{-1}$ to 0.3 g g$^{-1}$ in 3 hours (Pagonis et al., 2023). While Garofalo et al. found no significant change of NEMRs between 0.5-8 hours transport of smoke from 20 western wildfires, they concluded that there was secondary OA formation through oxidation driven condensation, but it was balanced by dilution-driven evaporation (Garofalo et al., 2019). Gkatzelis et al. reported the NEMRs of some plumes that were more than an hour old and are shown in Table S10 with their corresponding physical age (Gkatzelis et al., 2024). For the same fire (William's flat), the NEMR was 0.331 µg m$^{-3}$ ppb$^{-1}$ at a physical age of 15 minutes that increased to 0.524 µg m$^{-3}$ ppb$^{-1}$ at 102 minutes (Gkatzelis et al., 2024). Similar increase for the Castle fire was seen where the NEMRs reported are 0.204, 0.244, and 0.463 µg m$^{-3}$ ppb$^{-1}$ at 25, 27, and 153 minutes respectively. For another fire (Horsefly), the NEMR was 0.398 µg m$^{-3}$ ppb$^{-1}$ at a physical age of 65 minutes and remained at a similar value of 0.391 µg m$^{-3}$ ppb$^{-1}$ at 104 minutes. On average, the mean NEMRs for plumes of physical age less than one hour, reported in their study, was $0.218 \pm 0.110$. This value is lower than that of plumes older than one hour, which have a mean value of $0.391 \pm 0.131$ (Gkatzelis et al., 2024). Overall, we find no trends in our data when considering all the smoke plumes detected, but for periods of expected photochemical activity we observe consistent evidence for aerosol formation with plume age, which might be attributed to the optically thin smoke that allows photochemistry throughout the plume compared to large optically thick wildfires that leads to more complex photochemistry within the plume (Decker et al., 2021).

*L581-583: "Satellite images do not show any visible differences in vegetation between the forested areas burnt on and off the base. Additionally, no further information regarding the fuel types in the off-base lands could be obtained."*

*Please note what satellite imagery was used. Also, the authors could consult recent vegetation maps (e.g. LANDFIE, https://www.landfire.gov/ ) which may provide a more sophisticated assessment of the vegetation (e.g. forest type) and potential fuel loading (https://www.landfire.gov/fuel) than is possible from visual inspection of imagery by a non-expert.*

**Response:** While we found LANDFIRE a very useful tool to use, it didn't show any significant difference in vegetation in the studied area. This and the satellite imagery used previously were noted in the manuscript.

L581-583: A preliminary assessment using Google Earth satellite imagery and Landscape Fire and Resources Management Planning Tool (LANDFIRE, https://www.landfire.gov/) does not show any visible differences in vegetation between the forested areas burnt on and off the base.

*L604-612: It may be worth including the lab study of Selimovic et al. (2018) in this discussion, something for the authors to consider.*

**Response:** Thank you for your valuable suggestion. The revised discussion lines are now as follows:

L604-612: Optical properties of absorptive aerosol spectral properties characterized by AAE are shown in Fig. 11 as a function of age. Total absorption AAE values from the two trailers with 7-wavelength aethalometers (i.e., BC+BrC measured by the aethalometer) varied between 1.31 and 3.32 (mean ± stdev of 1.89 ± 0.23) and between 3.19 and 7.43 (mean = 5.00 ± 0.89) for BrC only. AAEs have no trend with age for either fresh smoke plumes or periods of $O_3$ enhancement. While our total AAE values are similar (Zeng et al., 2022; Strand et al., 2016; Marsavin et al., 2023) or sometimes lower (Liu et al., 2016; Forrister et al., 2015) than those in other biomass burning studies, it is indicative of the presence of BrC in the smoke plumes studied. As for BrC AAEs, our reported values are significantly higher than those reported for western wildfires, where BrC determined from the PAS had an AAE of 2.07 ± 1.01 (Zeng et al., 2022), indicating difference in BrC optical properties or with instrumentation, which needs further investigation. Selimovic et al. show that duff has the highest AAE of 7.13 (calculated from absorption data at 401 and 870 nm) when burnt and it is typically consumed more in wildfires than in prescribed fires. However, the variability in optical properties is influenced more by the differential consumption of individual components than by the dominant tree species in the ecosystem. (Selimovic et al., 2019).

*615-616: "The method was successful in capturing a significant number of smoke events (64)."*

*Include the number of days on which smoke events were successfully measured.*

**Response:** The manuscript was edited as recommended.

L615-616: The method was successful in capturing a significant number of smoke events (64) monitored on 42 days across 2 burning seasons.

*617: change 'determined' to 'estimated' (the two methods used at times yielded very different smoke age estimates)*

**Response:** The manuscript was edited as recommended.

L617: Source and age for each smoke plume detected was  estimated.

*L617-618: Would be useful to include the number of fires for which smoke events were measured.*

**Response:** The number of fires was added to L615-616 as follows:

L615-616: The method was successful in capturing a significant number of smoke events (64) monitored on 42 days and linked to 45 fires across 2 burning seasons.

***Figures***

*Maps need scale bars!*

*Figure 1. Nice figure, but please add scale bar.*

**Response:** Thank you for pointing this out. A scale bar was added to Figure 1 (a and b) and Figure 2a.

*Figure 3. Regression line in missing in panel b.*

**Response:** The regression line was added in panel b of Figure 3.

*Figure 10. Since wind vector and hysplit are on same scale (as they should be) maybe use just one set of labels (smoke age) for the x-axis. Is the ozone enhancement plotted versus wind vector age or hysplit age? Please clarify in caption.*

 **Response:** One set of labels was removed, and caption was edited as follows:

[Figure]

**Figure 3.** (a) $PM_{2.5}$ mass, (b) BC, and (c) BrC NEMRs of all studied smoke events as a function of age estimated using average wind vector and HYSPLIT analysis. Smoke plumes with observed $O_3$ enhancements are identified and plotted versus wind vector age. Linear regression coefficients of variation ($r^2$) for all data and for just $O_3$ enhancement periods are identified. Exponential fit equation for $PM_{2.5}$ mass NEMRs for $O_3$ enhancement periods is shown in (a).

*Fig. S6. Was all or most of the area within the burn units (gray shaded polygons) burned? Please add a scale bar to 6a. What is the native and displayed resolution of the Colombus airport and PCSG school monitoring sites? In tables S3 & S7, please state the reason for the missing CO observations.*

**Response:**

The shaded grey polygons correspond to the total area of the burn units, but the actual burned areas are typically a portion of the entire unit. While we don't have the exact boundaries of the burned areas, we have the total acres and the burnt acres values.

The displayed resolution is 1 hour but the information about the site does not clarify if measurements were taken at higher resolution.

The scale bar was added to Figure S6a and the reason for the missing CO observations was stated in each of the two indicated tables.

*Table S8. Please clarify in table heading that listed overpass times are local time.*

**Response:** Table heading was edited to:

**Table S8.** Satellite overpasses (local time) during the three smoke episodes shown in Figure 5.

*Technical:*

*L132: suggested text change to: "used as a tracer of smoke movement and dispersion". Also, it may be worth noting that CO is used as a standard tracer of combustion sources in atmospheric chemistry studies.*

**Response:** The sentence is revised:

L132: Carbon monoxide serves as a standard tracer for combustion sources in atmospheric chemistry studies since it is a relatively long-lived species, with a typical lifetime of $\sim 1$ month, emitted during incomplete combustion and used as a  tracer  of smoke movement and dispersion (Forrister et al., 2015; Liu et al., 2016).

*L334: "from the northwest, west, and southwest" change to "westerly" ?*

**Response:** The indicated line was edited in the manuscript as suggested.

*L614-615: "We describe a ground-based measurement method for characterizing smoke from prescribed fires based on continuous monitoring at multiple sites for an extended period in a regularly burned region."*

*Suggest changing 'ground-based measurement method' to 'ground-based observational study'*

**Response:** The sentence is revised into:

We describe a ground-based observational study  for characterizing smoke from prescribed fires based on continuous monitoring at multiple sites for an extended period in a regularly burned region. We focus on burning within a large military Fort in the southeastern US and identify the sources of the smoke to determine if it was within or outside the Fort and study emissions and evolution of species smoke species.

*L620: maybe change "conservative parameter" to "conserved co-emitted species"?*

 **Response:** The phrase was changed in the manuscript.

*L629: "indicating combined secondary O3 and particle mass formation" change to "indicating the formation of O3 and secondary organic aerosol"*

 **Response:** The phrase was changed in the manuscript:

L629: However, PM$_{2.5}$ mass NEMRs did increase with age for smoke detected in the afternoon in plumes where O$_3$ enhancements were observed, indicating the  formation of O$_3$ and secondary  aerosol.

**Other changes made in the manuscript, not directly related to any reviewer comments, are minor adjustments aimed at enhancing the clarity and structure of our sentences.**